MCNET-25-12

# Accelerating multijet-merged event generation with neural network matrix element surrogates

Tim Herrmann[1*], Timo Janßen[2,3], Mathis Schenker[1], Steffen Schumann[2], Frank Siegert[1]

**1** Institut für Kern und Teilchenphysik, TU Dresden, Dresden, Germany
**2** Institut für Theoretische Physik, Georg-August-Universität Göttingen, Göttingen, Germany
**3** Campus-Institut Data Science, Georg-August-Universität Göttingen, Göttingen, Germany

\* tim.herrmann@tu-dresden.de

## Abstract

The efficient simulation of multijet final states presents a serious computational task for analyses of LHC data and will be even more so at the HL-LHC. We here discuss means to accelerate the generation of unweighted events based on a two-stage rejection-sampling algorithm that employs neural-network surrogates for unweighting the hard-process matrix elements. To this end, we generalise the previously proposed algorithm based on factorisation-aware neural networks to the case of multijet merging at tree-level accuracy. We thereby account for several non-trivial aspects of realistic event-simulation setups, including biased phase-space sampling, partial unweighting, and the mapping of partonic subprocesses. We apply our methods to the production of Z+jets final states at the HL-LHC using the Sherpa event generator, including matrix elements with up to six final-state partons. When using neural-network surrogates for the dominant Z+5 jets and Z+6 jets partonic processes, we find a reduction in the total event-generation time by more than a factor of 10 compared to baseline Sherpa.

# 1  Introduction

The development of precise simulation methods is of utmost importance for the analysis and interpretation of data taken at current and future particle collider experiments. The combination of accurate theoretical predictions at the particle level provided by Monte Carlo event generators [1, 2] with detailed detector simulations that model the response of incident particles with the detector material [3–6] reflects our current understanding of particle production and interaction mechanisms. Such simulated data are used as digital twins of actual scattering events and are vital for the planning, optimisation, implementation, and interpretation of measurements in high-energy physics. Already for the ongoing experiments at the LHC and in particular for its high-luminosity upgrade it becomes computationally more and more challenging to provide sufficient amounts of simulated data. This is, on the one hand, due to the huge numbers of events needed to meet the statistical accuracy of (HL-)LHC measurements, on the other hand, ever more complex production processes and corresponding final states have to be addressed. Given the factorised nature of the simulation chain, we here focus solely on particle-level event generation.

Improving the resource efficiency of Monte Carlo generators is vital in order to meet the above-mentioned challenges. The computationally most expensive part of the event simulation is typically the computation of the hard-scattering matrix element, i.e. the evaluation of the squared transition amplitudes to a given perturbative order. For higher-multiplicity processes, these calculations get accomplished by dedicated matrix-element generators such as AMEGIC [7], COMIX [8], HELAC-PHEGAS [9, 10], MADGRAPH [11, 12], or WHIZARD [13]. In addition to the expressions for the respective matrix elements, these tools also construct suitable phase-space integrators that employ physics information about the process under scrutiny to devise optimised samplers. This offers two avenues for performance improvements beyond the current state-of-the-art. The first directly aims at a reduction of the amplitude evaluation times by finding their most compact representation [14] or by porting their evaluations to GPUs or vector CPUs [15, 16]. The second targets a reduction in the number of calls needed to reach a certain statistical accuracy when integrating, or, when generating unweighted events, in the rejection-sampling step. To reduce the variance of the underlying cross-section integration, novel sampling algorithms employing machine-learning techniques [17–26], or Markov Chain Monte Carlo type methods [27–29] are currently being developed. In particular for applications in experimental analyses, unit-weight events distributed according to the underlying production cross section are needed. To this end, weighted samples from the target distribution undergo a von-Neumann accept–reject step with respect to a predetermined maximum. The efficiency of this unweighting procedure can become rather small, in particular for high-multiplicity processes [30], for which the evaluation of the matrix element is, however, very costly. In Ref. [31] a novel two-step unweighting algorithm has been proposed that employs a fast neural-network surrogate to approximate the true event weight in a first unweighting step. In a second step events accepted based on their surrogate weight undergo another accept–reject procedure, effectively correcting for the mismatch with the actual event weight. This algorithm has been shown to yield unbiased iid event samples which are statistically equivalent to samples generated with the standard algorithm. In a follow-up study [32] improved network architectures, reflecting the QCD factorisation properties for rather soft and collinear kinematics [33], have been considered and shown to further boost the performance of the algorithm. For the single partonic channels contributing to $Z + 4, 5$ jets and $t\bar{t} + 3, 4$ jets production at the LHC that were considered, speed-up factors between 16 and 350 have been achieved for an implementation in the SHERPA event-generator framework [34, 35], based on tree-level matrix elements from AMEGIC [7].

Other neural-network methods for matrix-element regression that could alternatively be used as surrogates have for example been presented in Refs. [36–41].

In this publication we significantly extend the method towards fully realistic event samples as used by the experimental collaborations, based on merging parton-shower evolved processes of increasing jet multiplicity. To this end, a multitude of partonic channels needs to be considered, subjected to Sudakov suppression when matching with the parton shower. Furthermore, we include the effect of biased phase-space sampling aiming for an enhanced production of rare event kinematics, in particular in high-energy tails. We devise updated performance measures that provide the means to minimise the computational resources needed to generate a full particle-level sample. As a realistic application, we consider inclusive $Z$+jets production in $pp$ collisions at the LHC. We thereby include exact tree-level matrix elements for up to six jets provided by COMIX, which are significantly faster than the corresponding expressions from AMEGIC.

In Sec. 2 we briefly review the surrogate unweighting algorithm and discuss aspects related to its application in realistic event-generation setups. This includes the treatment for unobserved degrees of freedom, in particular the colour assignment of external states, and the biased generation of phase-space configurations in order to enhance certain kinematics, as well as the possible mapping of partonic channels. In Sec. 3 we introduce measures to quantify the performance gain when using surrogates. We further describe our choices for the hyperparameters of our networks and the strategy used to train them. Finally, in Sec. 4 we present results for the application of surrogate unweighting for inclusive $Z$+jets production at the HL-LHC based on the merging of tree-level matrix elements for up to six final state QCD partons. We carefully compile estimates for projected resource needs and compare to those for the conventional unweighting approach available in SHERPA. We present our conclusions and a brief outlook in Sec. 5.

## 2  Surrogates meet advanced Monte Carlo methods

The main motivation for our surrogate unweighting approach is to facilitate precise simulations for the HL-LHC experiments. Accordingly, it needs to be implemented within realistic HL-LHC calculational setups. These simulations employ not only highly accurate perturbative calculations, but also many algorithmic features of Monte Carlo event generators that make the generated event samples more practically usable or even just feasible. State-of-the-art simulation samples for the CPU-intensive bulk processes like $V$+jets or $t\bar{t}$+jets production make use of multijet merging at leading order (LO) or next-to-leading order (NLO), where matrix-element calculations of different jet multiplicity are combined with each other and with the parton shower to improve the description of QCD radiation. Especially for hard and wide-angle jets, fixed-order matrix elements for jet production provide a more reliable prediction than the soft-collinear approximation the parton shower is based on. At the same time, it is important to preserve the logarithmic accuracy of the QCD evolution in regions where resummation becomes important.

In this section, we study the non-trivial interplay of surrogate unweighting with such advanced methods, beyond the rather idealised setups so far used to explore the concepts of surrogate-based unweighting [31, 32]. The structure of the section is as follows: In Sec. 2.1 we review the original surrogate unweighting algorithm for individual partonic subprocesses. We continue with a general description of the multijet-merging method in Sec. 2.2. In the following subsections, we describe different advanced aspects of the event generation and their interplay with ME surrogates in each case. We begin with the integration over the colour degrees of freedom of external QCD particles in Sec. 2.3. Next, in

Sec. 2.4, we describe the options in SHERPA to bias the sampler towards certain kinematics or processes. Sec. 2.5 deals with partial unweighting, i.e. the controlled reduction of the weight maximum used to determine the acceptance probability in the unweighting step, and the overweight events associated with that procedure. Another algorithmic aspect is the taming of the combinatorial challenge in the number of contributing partonic channels, which is achieved by the mapping of channels with similar particle-flavour structure, as described in Sec. 2.6. Finally, we consider the sampling over the individual subprocesses contributing to a multijet-merged calculation in Sec. 2.7. In Sec. 3, we will already exemplify some of the improvements using a realistic LHC event generation setup for $Z$+jets production. A detailed description of the computational setup will be provided in Sec. 4, where we quantitatively assess the performance improvements that can be achieved and confirm that our surrogate unweighting method can lead to significant gains also in a realistic multijet-merged setup for the HL-LHC.

## 2.1 Original surrogate unweighting algorithm

The differential hadronic cross section for each scattering event contains various contributions, each of which contributes to the total event weight, and is defined as

$$
\mathrm{d}\sigma_{ab\to n}|_{p_a,p_b,\{p_i\}} = \underbrace{f_a(x_a,\mu_F)\, f_b(x_b,\mu_F)}_{w_{\mathrm{PDF}}} \underbrace{|\mathcal{M}_{ab\to n}|^2}_{w_{\mathrm{ME}}} \underbrace{|J_{\Phi_n}|}_{w_{\mathrm{PS}}} \Theta_n(Q_{\mathrm{cut}})\, \mathrm{d}x_a\, \mathrm{d}x_b\, \mathrm{d}\Phi_n|_{p_a,p_b,\{p_i\}}\,,
$$
(1)

where $\Theta_n(Q_{\mathrm{cut}})$ implements phase-space cuts and constraints on the core process, and demands at least $n$ resolved jets above the jet-resolution scale $Q_{\mathrm{cut}}$ (for definitions of the core process and the jet-resolution scale, see Sec. 2.2). The parton density functions (PDFs) for the colliding hadrons get combined into the weight factor $w_{\mathrm{PDF}}$, the squared matrix element yields $w_{\mathrm{ME}}$, and possible Jacobian factors from the parametrisation of the phase-space element $dx_a dx_b\, \mathrm{d}\Phi_n$ are summarised in $w_{\mathrm{PS}}$. In principle, all parts of the calculation might be substituted with machine-learned surrogates $s$. However, following Ref. [32], we here focus on the squared QFT scattering matrix element directly, seeking an approximation $s_{\mathrm{ME}} \approx w_{\mathrm{ME}}$, and will not include the parton density functions nor the phase-space factors in the training of our networks.

Given a well-trained surrogate for the matrix-element weight, the two-stage unweighting algorithm proceeds as follows:

1. For a given phase-space point, the matrix-element surrogate $s_{\mathrm{ME}}$, as well as the phase-space and PDF weights, are evaluated. Their product yields the surrogate event weight $s$. The point is then accepted with probability

$$
p_{\mathrm{1st,surr}} = \min(1, s/w_{\mathrm{max}})\,,
$$
(2)

where $w_{\mathrm{max}}$ denotes a predetermined maximal event weight.

2. Only for events passing the first unweighting step, the full matrix element gets evaluated, producing the true event weight $w$. The ratio

$$
x \equiv w/s
$$
(3)

is computed and determines the final acceptance probability of the event, given by

$$
p_{\mathrm{2nd,surr}} = \min(1, x/x_{\mathrm{max}})\,.
$$
(4)

Similar to $w_{max}$, the overestimate $x_{max}$ needs to be predetermined. The event is ultimately assigned the weight

$$\widetilde{w} = \max(1, s/w_{max}) \cdot \max(1, x/x_{max}) \,. \tag{5}$$

This results in partially unweighted samples where some events carry overweights.

We motivate the use of partial unweighting in Sec. 2.5, where we also explain how to deal with overweight events, especially in the context of surrogate unweighting.

## 2.2 The multijet-merging method

We want to remind the reader of the key ingredients of the multijet-merging method [42, 43], as implemented in SHERPA, which are relevant to incorporate surrogate unweighting. The generic multijet-merged cross section is build-up from processes of increasing parton multiplicity:

$$d\sigma = \sum_{n=0}^{n_{max}-1} d\sigma_n^{excl} + d\sigma_{n_{max}}^{incl} \,. \tag{6}$$

Each exclusive $n$-jet process—relative to some core process which may itself contain jets that do not get counted by $n$—can be evaluated at LO or NLO QCD. The notion of exclusive and inclusive cross sections refers to the shower evolution of the parton ensemble and is implemented through a jet-resolution parameter $Q_{cut}$, called the merging scale. Its role is to slice the emission phase space according to a hardness-measure $Q$ into a region $Q < Q_{cut}$, where emissions are produced by the parton shower, and $Q > Q_{cut}$, where emissions are produced from real-radiation matrix elements. This avoids a naive double counting of emission probabilities. The multijet matrix-element configurations get clustered using a "reverse parton shower" to probabilistically interpret how it would have emerged from shower evolution. This determines the shower starting conditions for subsequent emissions and enables the following two steps to reinstate the shower's logarithmic accuracy within the merged QCD evolution:

(i) Sudakov factors $\Delta(t, t') = \exp\left\{ -\int_t^{t'} K(\Phi_1) \, d\Phi_1 \right\}$, with shower-splitting kernels $K$ that get integrated over the single-emission phase space $\Phi_1$ [42], are applied on internal and external legs of the clustered configuration using a truncated shower and corresponding vetoes (yielding an average efficiency $\epsilon_{Sudakov}$).

(ii) The effective renormalisation scale $\mu_R$ is thereby defined as a combination of the core process scale $\mu_{R,core}$ (to order $n_{core}$) and the scale of each (shower or matrix element) emission, $p_{T,i}$, as

$$\alpha_s^{n_{core}+k}(\mu_R^2) \equiv \alpha_s^{n_{core}}(\mu_{R,core}^2) \prod_{i=1}^{k} \alpha_s(p_{T,i}^2) \,. \tag{7}$$

We note that the Sudakov veto efficiency $\epsilon_{Sudakov}$, as well as the time needed to accomplish the full shower evolution of an event depends on the partonic channel and the very event kinematics, i.e. the phase-space region probed.

## 2.3 Integrating over colour degrees of freedom

The matrix elements evaluated in Eq. (1) include an implicit sum over unobserved degrees of freedom. This includes in particular the $SU(3)_c$ colour-charge assignments of external

QCD particles. In modern matrix-element generators there exist two ways of treating those, either through an explicit sum over all non-vanishing colour configurations, or, alternatively, through sampling them probabilistically. In particular for processes of high jet-multiplicity, colour sampling will often outperform colour summation [14], despite the high dimensionality of the colour space to explore and the corresponding reduced unweighting efficiency.

In our previous studies [31, 32], surrogates were derived for matrix elements from the AMEGIC generator [7], which only implements the colour-summed option. We here instead consider matrix elements from COMIX [8], which in its latest release included in SHERPA-3.0 supports both options, which allows us to directly compare the performance for both colour treatments. Since the default and most widely used option is colour-sampling, we denote this as "Baseline" in the following comparisons, while the colour-summed one is denoted as "Baseline$_\Sigma$".

**Interplay with ME surrogates**   There are two aspects where the treatment of QCD colour affects our surrogate approach. First, the neural-network architecture we use to approximate QCD matrix elements is based on Catani–Seymour dipole factorisation [44]. We thereby employ colour-summed dipole functions [32, 33], which are well-suited for colour-summed matrix elements. However, when considering colour sampling, the performance of the surrogate in approximating the matrix element deteriorates significantly, eventually becoming prohibitively low [32]. In what follows we will therefore apply surrogate unweighting only to colour-summed matrix elements. To alleviate this issue, coloured dipole expressions should be used in the neural-network surrogate which we leave for future work, though.

The second aspect of the colour treatment that affects the target function of the surrogates is related to the actual evaluation times of the matrix elements. In fact, the ratio of matrix-element to phase-space computation time is very unfavourable for the surrogate ansatz when using colour-sampled matrix elements. So far, the central figure of merit of our surrogate unweighting method, the effective CPU gain factor, has been quoted in reference to colour-summed computations. However, as will be shown in Sec. 4, for high-multiplicity processes the relevant benchmark might rather be the colour-sampled result, which needs to be accounted for when claiming performance gains due to the usage of surrogates. Furthermore, COMIX's phase-space sampler typically achieves a higher unweighting efficiency than the one used by AMEGIC, reducing the theoretically possible effective gain factor by 1-2 orders of magnitude.

## 2.4   Phase-space biasing

Addressing the entire LHC physics programme with corresponding Monte Carlo simulations presents a major computing challenge. This includes high-statistics samples for precision measurements of standard-candle processes, as well as predictions for rare event kinematics, selected through kinematic cuts, e.g. by demanding a certain number of high-$p_T$ jets, when seeking for new-physics signals. To cover the diversity of specific analyses, focusing on different phase-space regions, slicing the Monte Carlo production for a specific process into separate disjoint samples is a standard procedure; see, for example, [45]. However, this approach entails several caveats in terms of practicability: a multitude of specialised samples need to be handled consistently, the statistical coverage of phase space is rather discontinuous, and possibly large event-weight discontinuities can show up when transitions across slice boundaries appear in later stages of the simulation.

An attractive alternative to phase-space slicing is provided by a continuous phase-space biasing, i.e. the controlled over- or underproduction of certain kinematics or processes. In

SHERPA, phase-space biasing is possible at two levels:

- at process level, the contribution of a given process or final-state multiplicity can be artificially increased by an enhancement factor[1],

- at event level, the contribution of certain phase-space configurations $\Phi$ can be artificially increased by a function $h(\Phi)$, defined by the user via

  - an explicit enhancement function $h(\Phi)$, given in terms of final-state momenta,
  - an enhancement observable $O(\Phi)$ in which events should be distributed uniformly, i.e. phrased in terms of an enhancement function:

$$h(\Phi) \equiv \frac{1}{\frac{d\sigma}{dO}\big|_{O(\Phi)}} \,. \tag{8}$$

The raw event weight for a given phase-space point $\Phi$ used in the unweighting procedure then becomes:

$$w \equiv w_{\mathrm{PDF}} \cdot w_{\mathrm{ME}} \cdot w_{\mathrm{PS}} \cdot h \,. \tag{9}$$

To enhance rare high-$p_T$ events and multijet events in our $Z$+jets simulations throughout this work, the following enhancement function is used for subprocesses with at least one final-state parton $P$:

$$h(\Phi) = \left( \max\left( \frac{\sum_{i \in P} p_{T,i}}{20\,\mathrm{GeV}}, \frac{p_{T,e^+e^-}}{20\,\mathrm{GeV}} \right) \right)^2 \,, \tag{10}$$

with $p_{T,i}$ and $p_{T,e^+e^-}$ the transverse momenta of the final-state partons and the dilepton pair, respectively. This functional form is inspired by the typical statistical biasing used in LHC experiments, see, for example, Ref. [45]. Given this is in general a non-trivial function of the final-state parton multiplicity, it will induce a corresponding enhancement in that as well [2]. This effect is demonstrated in Table 1, where we combine the mean enhancement weights of each subprocess, $\langle h \rangle_{\mathrm{subprocess}} \equiv \frac{\langle h \cdot w/h \rangle}{\langle w/h \rangle}$, into a weighted average for the whole group of processes, i.e. for processes of given jet multiplicity, defined as:

$$\langle h \rangle_{\mathrm{group}} \equiv \frac{\langle \langle h \rangle_{\mathrm{subprocess}} \cdot \sigma_{\mathrm{subprocess}} \rangle}{\langle \sigma_{\mathrm{subprocess}} \rangle} \,. \tag{11}$$

We observe that upon application of the enhancement function, the spread in the original $Z+N$-jet cross sections is significantly reduced. For example, while the plain $Z+6$-jet cross section is a factor 185 smaller than that for $Z + 1$-jet production, with the enhancement according to Eq. (10) they are basically on par. Note that while these enhancements and cross sections are representative of the sample composition at the parton level, this might not be the case when invoking the parton shower in a multijet-merged sample. The shower Sudakov factors appearing there are phase-space dependent and might alter this picture by suppressing previously enhanced regions of phase space.

---

[1]This option is not used in the comparisons in this work and thus dropped from the equations for simplicity, but straightforward to include also in the context of surrogate unweighting.

[2]In fact, the normalisation of 20 GeV is chosen here such that the global enhancement of the $\geq 1$-jet process with respect to the 0-jet one reaches a desired given value.

| Jet Multiplicity | Enhancement | Cross section | |
|---|---|---|---|
| | | nominal | enhanced |
| $N$ | $\langle h \rangle_{\text{group}}$ | $\sigma_{\text{group}}$ [pb] | $\langle h \rangle_{\text{group}} \cdot \sigma_{\text{group}}$ [pb] |
| 0 | 1.0 | 1 793.2(7) | 1 793.2(7) |
| 1 | 5.7 | 650.0(9) | 3 705(5) |
| 2 | 39.3 | 248.3(6) | 9 758(24) |
| 3 | 131 | 96.4(4) | 12 628(52) |
| 4 | 315 | 34.8(2) | 10 962(63) |
| 5 | 647 | 11.2(1) | 7 246(65) |
| 6 | 1 191 | 3.5(1) | 4 169(119) |

Table 1: Mean enhancement factor $\langle h \rangle_{\text{group}}$ and cross section $\sigma_{\text{group}}$ for the different jet multiplicities of $pp \rightarrow Z + N$-jet production at the LHC. See Sec. 4 for details on the event selections.

**Interplay with ME surrogates**   Given that we construct surrogates from $w_{\text{ME}}$, excluding the enhancement $h$, our networks should nominally be independent of the enhancement itself. Nevertheless, there are several aspects that result in a non-trivial interplay:

1. The differential enhancement weights affect the weight distribution and thus the efficiencies in accurately predicting the event weight. If Mean Squared Error (MSE) on $w_{\text{ME}}$ is used as loss function during network training, which is not sensitive to the efficiency, there is no enhancement dependence during training. However, if one uses directly the gain factor as loss function (cf. Sec. 3.2), some dependence of the networks on the enhancements appears via the regression quality, i.e. the efficiencies.

2. Since surrogates might be more or less efficient for different regions in phase space, also the decision whether to use them for a given subprocess depends on the enhancement used.

3. The assessment of gain factors of the surrogate unweighting approach below will only be realistic for LHC event generation if done in the presence of a phase-space enhancement that mimics the needs of typical LHC experiments.

## 2.5   Maximum reduction and overweight events

The usage of weighted events within LHC experimental analyses is practically not affordable, given that each event, independent of its actual contribution to the cross section, needs to be propagated through a time-consuming detector simulation. Accordingly, to maximise the statistical power of an average Monte Carlo simulated event, unweighted events are used per default. However, a perfect unweighting with the actual maximum of the differential cross section in the considered phase space is often neither feasible nor necessary. Instead, a "partial unweighting" approach is used that relies on a reduced maximum $w_{\text{max}}$, implicitly defined as the lowest weight $w$ in the set of all weights $\mathbf{w}$ satisfying

$$\sum_{w \in \mathbf{w}, w > w_{\text{max}}} w \leq \epsilon_{\text{max}} \sum_{w \in \mathbf{w}} w \,. \tag{12}$$

This introduces a new steering parameter $\epsilon_{\text{max}} \in [0, 1]$, which can systematically be optimised for specific application scenarios [46].

In the baseline approach, using the full matrix-element calculation without surrogates, a phase-space point gets accepted with probability

$$p_{\text{baseline}} \equiv \min\left(1, \frac{w}{w_{\max}}\right),$$ (13)

resulting in an averaged acceptance efficiency $\epsilon_{\text{baseline}}(w, w_{\max}) \equiv \langle p_{\text{baseline}} \rangle$ [3]. For $\epsilon_{\max} > 0$, some event weights might exceed the reduced maximum $w_{\max}$ during unweighting. In consequence, such events retain a non-unit weight, called overweight, even after unweighting:

$$w_{\text{overweight}} \equiv \max\left(1, \frac{w}{w_{\max}}\right).$$ (14)

Note that the overweight treatment described here is also used as a rescue system for the case $\epsilon_{\max} = 0$, if during event generation events overshoot the maximal weight value, determined in a finite-statistics pre-run, against which the rejection sampling is performed.

**Interplay with ME surrogates** Introducing a reduced weight maximum and correspondingly overweight events affect the performance of the baseline unweighting algorithm, as it leads to an increase of the acceptance probability. Consequently, the choice for $\epsilon_{\max}$ affects the potential time savings when using a surrogate and has an impact on the statistical significance of individual events.

The maximum reduction technique can also be used for the first and second unweighting step in the surrogate approach, i.e. in the initial step based on the weight approximation from the surrogate $s$, and when correcting the mismatch between the surrogate and the true event weight $w$. In the first unweighting step, in full analogy to $w_{\max}$, a reduced maximum for the surrogate weights, $s_{\max}$, can be defined in dependence on the parameter $\epsilon_s$, according to

$$\sum_{s \in \mathbf{s}, s > s_{\max}} s \leq \epsilon_s \sum_{s \in \mathbf{s}} s.$$ (15)

The correction factor used in the second step[4] reads

$$x \equiv \frac{w}{s} \cdot \max\left(1, \frac{s}{s_{\max}}\right).$$ (16)

The reduced maximum correction weight, $x_{\max}$, is implicitly given in dependence on $\epsilon_x$:

$$\sum_{x \in \mathbf{x}, x > x_{\max}} w \leq \epsilon_x \sum_{x \in \mathbf{x}} w.$$ (17)

For the first unweighting step, the average probability to keep an event is similar to the classical approach, but solely based on the surrogate:

$$\epsilon_{\text{1st,surr}}(s, s_{\max}) \equiv \left\langle \min\left(1, \frac{s}{s_{\max}}\right) \right\rangle.$$ (18)

The efficiency of the second unweighting is given by the probability that the event passes the second unweighting step, given that it passed the first stage. The condition to have passed the first unweighting can be included through a weighted average, i.e.

$$\epsilon_{\text{2nd,surr}}(x, x_{\max}) \equiv \left\langle \min\left(1, \frac{x}{x_{\max}}\right) \right\rangle_{\text{weight} = \min(s, s_{\max})}.$$ (19)

---

[3]This efficiency is estimated in Ref. [32] to be: $\epsilon_{\text{baseline}} \approx \frac{\langle w \rangle}{w_{\max}}$. This approximation is valid for rather small $\epsilon_{\max} \lesssim 0.01$ as used here. However, when considering the full range $\epsilon_{\max} \in [0, 1]$ (see Ref. [46]) the full expression needs to be used.

[4]The definition of $x$ used in this paper is different from the definition in Eq. (3) used in the original surrogate-unweighting algorithm, cf. also the discussion below.

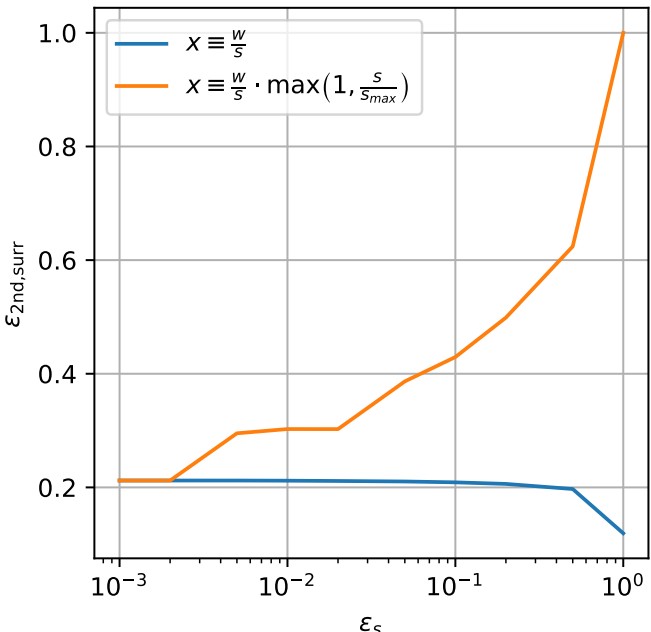

Figure 1: Comparison of the efficiency of the second unweighting step for the process $gu \to Zgggggu$ for two alternative definitions of the correction weight $x$ as described in the text. The resulting efficiencies are displayed as a function of the first unweighting maximum reduction, $\epsilon_s$. The blue line uses $\epsilon_x = 0.001$, while for comparability the orange line has an adjusted $\epsilon_x$, so that for each value of $\epsilon_s$ both curves feature the same contribution to the total cross section from overweights.

These expressions for the two unweighting efficiencies need to be used when exploring the full range of $\epsilon_s$ and $\epsilon_x$, see Ref. [46]. However, for small values of $\epsilon_s$ and $\epsilon_x$ they can be approximated to

$$\epsilon_{\text{1st,surr}} \approx \frac{\langle s \rangle}{s_{\max}} \,, \tag{20}$$

$$\epsilon_{\text{2nd,surr}} \approx \frac{\langle x \rangle_{\text{weight}=s}}{x_{\max}} \,. \tag{21}$$

The final probability of accepting an event in the two-stage surrogate unweighting is given by

$$p_{\text{surr}} \equiv \min\left(1, \frac{s}{s_{\max}}\right) \cdot \min\left(1, \frac{x}{x_{\max}}\right) \,, \tag{22}$$

with $s_{\max}$ and $x_{\max}$ the reduced maxima for the first and second unweighting step, leading to partial unweighting with possible overweights given by

$$x_{\text{overweight}} \equiv \max\left(1, \frac{x}{x_{\max}}\right) \,. \tag{23}$$

In contrast to the original version of the algorithm, described in [31, 32] and briefly summarised above in Eq. (3), the correction weight Eq. (16) here includes an additional factor from Eq. (14) as first proposed in [47]. Depending on the choices for $\epsilon_s$ and $\epsilon_x$, this can further improve the performance of the algorithm. As an example, in Fig. 1 we present the efficiency $\epsilon_{\text{2nd,surr}}$ of the second unweighting step as a function of the first unweighting maximum reduction $\epsilon_s$. Specifically, we consider the partonic channel $gu \to Zgggggu$ contributing to $Z + 6$-jet production, and we fix the second unweighting

| Jet Multiplicity $N$ | subprocesses | | reduced | |
|---|---|---|---|---|
| | all quarks | $\leq 4$ quarks | all quarks | $\leq 4$ quarks |
| 0 | 5 | | 2 | |
| 1 | 15 | | 6 | |
| 2 | 95 | | 30 | |
| 3 | 145 | | 50 | |
| 4 | 485 | 160 | 199 | 56 |
| 5 | 635 | 160 | 277 | 56 |
| 6 | 1595 | 160 | 836 | 56 |

Table 2: Number of partonic subprocesses contributing to hadronic $Z + N$-jet production (second column) and their reduced set when mapping initial- and final-state quark flavours. For $N \geq 4$, the numbers are also given for the case when limiting to at most 4 external quarks.

maximum reduction to $\epsilon_x = 0.001$ for the original algorithm, defined in Eq. (3), shown in blue. The $\epsilon_x$ of the improved algorithm, defined in Eq. (16), shown in orange, is adjusted such that it matches the overweight contribution of the original algorithm. So, for a comparable statistical power, we here ensure the same contribution to the total cross section from overweights in the two approaches for a given $\epsilon_s$. While the choice of $\epsilon_s$ primarily affects the efficiency of the first unweighting and the resulting fraction of overweights from that step, it also has an impact on the second unweighting efficiency as shown in Fig. 1: When keeping the second unweighting overweight contribution constant at $\epsilon_x = 0.001$, the total fraction of overweights increases with rising $\epsilon_s$ for the original algorithm. Since the improved algorithm can partly compensate the overweights from the first unweighting step, it reaches higher second unweighting efficiencies for the same total fraction of overweights. The improved algorithm makes the whole two step unweighting more efficient compared to the original algorithm, especially for larger values of $\epsilon_s$.

## 2.6 Process mapping

When considering processes of increasing jet multiplicity the proliferation in the number of contributing partonic channels is very rapid. This is illustrated in Tab. 2, where the second column quotes the number of distinct partonic subprocesses contributing to $Z + N$-jet production. The five distinct channels contributing to the 0-jet process are of the type $q\bar{q} \to Z$, where as quark flavours we consider $q \in \{u, d, c, s, b\}$, with $m_q = 0$. However, these five channels can easily be reduced to two generic channels, when mapping all up-type flavours onto the $u$-quark and all down-types onto the $d$-quark, and respectively for the anti-particles. The differences within these two groups are solely due to the PDFs. These, however, can be treated in a fully factorised manner. The partonic matrix elements for the two up-quark and the three down-quark initiated channels are otherwise identical. For the 1-jet process the six generic process groups read $u\bar{u} \to Zg$, $d\bar{d} \to Zg$, $ug \to Zu$, $dg \to Zd$, $\bar{u}g \to Z\bar{u}$, and $\bar{d}g \to Z\bar{d}$, where again $u \in \{u, c\}$ and $d \in \{d, s, b\}$. Similarly, for all processes contributing to $pp \to Z + N$-jets, a reduced set of basic channels can be identified, onto which all other processes of the respective multiplicity can be mapped. The identification of process mappings is fully automated in the SHERPA generator and used per default.

For actual high-multiplicity processes it is common practice to discard contributions

from channels with more than two quark lines, i.e. four external quarks [48][5]. Compared to adding an additional gluon, the splitting of a gluon into a $q\bar{q}$ pair is colour suppressed. In this approximation, the growth in the number of subprocesses saturates at $Z + 4$ jets with 160 channels. Taking into account initial- and final-state flavour symmetries this corresponds to a maximum of only 56 mapped channels that need to be considered. All other processes can identically be mapped on one of those.

**Interplay with ME surrogates**   In view of surrogate unweighting, an obvious consequence of mapping partonic processes is that neural-network emulators only need to be trained for the reduced set of generic process groups. Consequently, training effort and the amount of data needed can be significantly reduced. In fact, this allows us to use training data generated already during the initial integration phase of SHERPA, right after the integrators have been optimised. Given that the subprocess-specific PDF and phase-space weights contribute to the total weight in a factorised manner, these can be combined with the matrix-element weight of the corresponding process group during the event-generation phase, to obtain fully exclusive events. The PDF weight is thereby calculated jointly with the phase-space weight, resulting in the combined evaluation time

$$\langle t_{\mathrm{PS}} \rangle = \langle t_{\mathrm{PDF}} \rangle + \langle t_{\mathrm{PS, \ raw}} \rangle + \frac{1}{\epsilon_{\mathrm{cut}}} \langle t_{\mathrm{cut}} \rangle \,. \tag{24}$$

The contribution proportional to $\langle t_{\mathrm{cut}} \rangle$ measures the total time needed to compute phase-space constraints and to check if proposed points hit the fiducial phase space. Accordingly, $\epsilon_{\mathrm{cut}}$ measures the corresponding cut efficiency, which in particular for high-multiplicity processes with non-trivial phase-space cuts can be significantly smaller than unity. Note that in an earlier study [31], the inclusion of the phase-space weight was considered for a simplified surrogate neural network architecture. However, the dipole-factorisation ansatz from [33] used here is tailored to approximate the plain QCD matrix element. Although the PDF weight could easily be included in the surrogate, this would require an increase in the number of trained networks, given that surrogates for the full set of distinct initial-state flavour assignments would need to be learned.

To possibly further reduce the number of neural networks needed to provide surrogates for the full set of partonic processes contributing to $Z + N$-jet production, we explore the correlations between the matrix-element weights of different process groups for fixed final-state multiplicity. To this end, we evaluate the Pearson correlation coefficients for the true matrix-element weights $w_{\mathrm{ME}}$ of all pairs of distinct process-group channels of the same jet multiplicity. In Tab. 3 we list the Pearson correlations between selected process groups. In particular, we quote the highest and lowest correlations found between channels contributing to 5- and 6-jet production. For both multiplicities we find processes with correlations well above 0.9. These share the feature that the two considered processes have very similar, i.e. identical, initial states. In contrast, the correlations between the channels with $gq$ and $q\bar{q}$ initial states are rather low. However, note that here we rely on the ordering of external flavours as realised in SHERPA. The underlying QCD amplitudes indeed exhibit permutation symmetries that would in principle make it possible to achieve higher correlations upon reordering external legs accordingly. We furthermore show the pairwise correlations between the three processes with the largest run time, i.e. those computationally most expensive. These are all $Z + 6$-jet channels featuring a single quark line that stretches from the initial to the final state, with the external partons ordered in the same manner. The residual differences originate from the different $U(1)$ and $SU(2)$ charges of the involved quarks. Consequently, we observe sizeable correlation coefficients

---

[5]While in SHERPA this is not done per default, such limit can be invoked by setting `Max_N_Quarks = 4`.

| Process 1 | Process 2 | Correlation |
|---|---|---|
| $Z + 5$-jets highest and lowest | | |
| $d_1\overline{d}_1 \to e^+e^- g g g d_1\overline{d}_1$ | $u_1\overline{d}_2 \to e^+e^- g g g u_1\overline{d}_2$ | 0.99 |
| $g u_1 \to e^+e^- g g d_2 u_1\overline{d}_2$ | $u_1\overline{d}_2 \to e^+e^- g g g u_1\overline{d}_2$ | 0.001 |
| $Z + 6$-jets highest and lowest | | |
| $g g \to e^+e^- g g d_1 d_2\overline{d}_1\overline{d}_2$ | $g g \to e^+e^- g g u_1 d_2\overline{u}_1\overline{d}_2$ | 0.98 |
| $g\overline{u}_1 \to e^+e^- g g g d_2\overline{u}_1\overline{d}_2$ | $u_1\overline{u}_2 \to e^+e^- g g g g u_1\overline{u}_2$ | 0.001 |
| $Z + 6$-jets top 3 run time contribution | | |
| $g u_1 \to e^+e^- g g g g g u_1$ | $g d_1 \to e^+e^- g g g g g d_1$ | 0.95 |
| $g u_1 \to e^+e^- g g g g g u_1$ | $g\overline{d}_1 \to e^+e^- g g g g g\overline{d}_1$ | 0.83 |
| $g d_1 \to e^+e^- g g g g g d_1$ | $g\overline{d}_1 \to e^+e^- g g g g g\overline{d}_1$ | 0.80 |

Table 3: Selected Pearson correlation coefficients for the matrix-element weights $w_{\mathrm{ME}}$ for process groups contributing to $Z + 5, 6$-jet production. Shown are the highest and lowest correlations found for 5- and 6-jet final states, respectively. Furthermore, the correlations between the three processes with the highest overall run time are given.

between 0.8 and 0.95. While in this work we do not consider the use of a reduced basis of process-group channels, our findings clearly motivate to seek for a minimal basis of generic channels for which surrogates would need to be trained.

## 2.7 Sampling over multiple subprocesses

The Monte Carlo simulation of multijet-merged processes spans various individual processes, be it for different parton multiplicities, or, within the same parton multiplicity, for different parton combinations and flavours. During the generation of each event, one such exclusive process will have to be selected according to the process's cross section $\sigma$ and optionally a process enhancement $h$ as introduced in Sec. 2.4. Depending on the event-generation mode, this yields a selection probability of

$$p_{\mathrm{sel}} \equiv \frac{w_{\mathrm{sel}}}{\sum_{\mathrm{subprocesses}} w_{\mathrm{sel}}} \, , \tag{25}$$

where

$$w_{\mathrm{sel}} = \begin{cases} \frac{\sigma_{\mathrm{subprocess}} \cdot \langle h \rangle_{\mathrm{subprocess}}}{\epsilon_{\mathrm{cut}}} = \langle w \rangle & : \quad \text{weighted events} \, , \\ \frac{\sigma_{\mathrm{subprocess}} \cdot \langle h \rangle_{\mathrm{subprocess}}}{\epsilon_{\mathrm{baseline}} \cdot \epsilon_{\mathrm{cut}}} = w_{\max} \cdot \langle w_{\mathrm{overweight}} \rangle \approx w_{\max} & : \quad \text{unweighted events} \, . \end{cases} \tag{26}$$

For unweighted events, this contains the unweighting efficiency $\epsilon_{\mathrm{baseline}}$ in order to get the same relative number of events after the event rejection in the unweighting step. The final event weight after unweighting in the baseline approach is then constant except for corrections due to the phase-space biasing and the maximum reduction,

$$w_{\mathrm{final}}^{\mathrm{baseline}} \equiv \frac{1}{h} \frac{w_{\max} \cdot w_{\mathrm{overweight}}}{p_{\mathrm{sel}}} = \frac{1}{h} \frac{w_{\mathrm{overweight}}}{\langle w_{\mathrm{overweight}} \rangle} \cdot \sum_{\mathrm{subprocesses}} w_{\mathrm{sel}} \, . \tag{27}$$

Given it presents the default used in SHERPA, the approximation in Eq. (26) is also employed here.

**Interplay with ME surrogates** In order to correctly take into account the efficiency of the surrogate-unweighting method, the selection probability has to be adjusted to

$$w_{\text{sel}} = \frac{\sigma_{\text{subprocess}} \cdot \langle h \rangle_{\text{subprocess}}}{\epsilon_{\text{1st,surr}} \cdot \epsilon_{\text{2nd,surr}} \cdot \epsilon_{\text{cut}} \cdot \alpha} \approx \frac{s_{\max} \cdot x_{\max}}{\langle x \rangle_{\text{weight}=s} \cdot \alpha} \, . \tag{28}$$

We include here an additional factor $\alpha$ that accounts for the statistical dilution due to the appearance of overweights, see Sec. 4.1.1. In this way, we ensure the same relative statistical information for each process. In principle, one could also add this term for the baseline approach in Eq. (26), but this is not done here, as there is usually nearly no statistical dilution for the common choice of $\epsilon_{\max} = 0.001$. The final event weight after (partial) surrogate unweighting is then given by

$$w_{\text{final}}^{\text{surrogate}} \equiv \frac{1}{h} \frac{s_{\max} \cdot x_{\max} \cdot x_{\text{overweight}}}{p_{\text{sel}}} \, . \tag{29}$$

From here on, the non-approximated form of Eq. (28) is used as there is no benefit in using the approximation.

# 3 DNN optimisation, training and evaluation

In this section we describe our approach towards a realistic performance measurement of our unweighting method using DNNs. Given the complexity of the event-simulation chain for a multijet-merged calculation in combination with a detector simulation, this measurement can become quite involved and depends on many parameters that can, in principle, be optimised.

To this end we define timing measures for the performance of unweighted event generation in Sec. 3.1. These serve as ingredients to our figure of merit, the effective gain factor of the surrogate in comparison to the baseline unweighting approach. The following subsections focus on the neural networks used as surrogates. There we describe our training method and hyperparameter choices.

The basic architecture of our neural networks has been described in Refs. [33, 49] and is based on the factorisation properties of QCD real-emission matrix elements in the soft and collinear limit:

$$|\mathcal{M}_{n+1}|^2 \to |\mathcal{M}_n|^2 \otimes \mathbf{V}_{ijk} \, . \tag{30}$$

The complicated singularity structure of this matrix element would be difficult to learn in a naive neural network, but is fully encapsulated in the universal dipole functions $\mathbf{V}_{ijk}$ [44, 50]. The factorisation-aware networks thus combine trained layers for the non-universal parts of multijet matrix elements with the analytically calculated dipole functions to achieve a highly accurate prediction of the matrix-element value. The input is thereby given by the 4-momenta of the incoming and outgoing particles, as well as kinematic variables formed thereof.

While the network architecture is carried over from our previous work, the network training is further improved here by explicitly using the CPU gain factor as target for the gradient-based optimisation as described in Sec. 3.2. In Sec. 3.3 we present our systematic optimisation of the remaining network hyperparameters, which we have done for the first time. These enhancements yield a significantly increased gain factor for small $\epsilon_x$, as demonstrated in Secs. 3.4–3.5, see also Appendix B.

## 3.1  Performance measures

The determination of the average time needed to generate a single (partially) unweighted event is rather straightforward. For the baseline approaches this is given by

$$t_{\text{unw}}^{\text{baseline}} \equiv \frac{\langle t_{\text{ME}} \rangle + \langle t_{\text{PS}} \rangle}{\epsilon_{\text{baseline}}} \,, \tag{31}$$

where $\langle t_{\text{ME}} \rangle$ is the average time needed for the matrix-element evaluation and $\langle t_{\text{PS}} \rangle$ is the average time needed for the phase-space generation (constructing the momenta, checking against phase-space cuts, evaluating the Jacobian, and, for hadron collisions, computing the PDFs). When using the surrogate approach, the two unweighting steps need to be taken into account separately, leading to

$$t_{\text{unw}}^{\text{surr}} \equiv \frac{\langle t_{\text{surr}} \rangle + \langle t_{\text{PS}} \rangle}{\epsilon_{\text{1st,surr}} \epsilon_{\text{2nd,surr}}} + \frac{\langle t_{\text{ME}} \rangle}{\epsilon_{\text{2nd,surr}}} \,, \tag{32}$$

where $\langle t_{\text{surr}} \rangle$ is the average time needed to evaluate the surrogate, and $\epsilon_{\text{1st,surr}}$ and $\epsilon_{\text{2nd,surr}}$ are the acceptance probabilities for the first and second unweighting step, respectively. In the second summand we take into account that the PDF and phase-space weights for the event have already been computed.

It is important to keep in mind that both times are on the same footing, i.e. the surrogate timing already includes the correction to the exact result in the second unweighting. No physics approximations are made in either of the approaches. Consequently, to quantify the benefit of using surrogate unweighting with respect to the colour-summed baseline approach, we can define an effective gain factor simply as the ratio of the two time measures,

$$f_{\text{eff}} \equiv \frac{t_{\text{unw}}^{\text{baseline}_\Sigma}}{t_{\text{unw}}^{\text{surr}}} = \frac{1}{\frac{\langle t_{\text{surr}} \rangle + \langle t_{\text{PS}} \rangle}{\langle t_{\text{ME}} \rangle + \langle t_{\text{PS}} \rangle} \cdot \frac{\epsilon_{\text{baseline}}}{\epsilon_{\text{1st,surr}} \epsilon_{\text{2nd,surr}}} + \frac{\langle t_{\text{ME}} \rangle}{\langle t_{\text{ME}} \rangle + \langle t_{\text{PS}} \rangle} \cdot \frac{\epsilon_{\text{baseline}}}{\epsilon_{\text{2nd,surr}}}} \,. \tag{33}$$

Generically, for a large gain we need to seek for a fast surrogate, i.e. low $\langle t_{\text{surr}} \rangle$, and a sizeable efficiency for the second unweighting step. This first definition of a gain factor will be the basis for our DNN training as described below, but will be refined later as performance measure in a realistic LHC event generation scenario in Sec. 4.1.

## 3.2  Gain factor as loss function

In Refs. [33, 49] we have trained our surrogate models in terms of a regression using a mean squared error (MSE) loss function. This works in practice, since a surrogate that approximates the true weights well will achieve a high efficiency in the two-step unweighting. However, the regression is ultimately only a proxy for our actual target: a large effective gain factor. Below, we use the definition of the gain factor to derive a loss function that can be used for neural-network training and that provides us with a more direct handle on the achieved gain of using surrogates in event unweighting.

We start from the definition of the effective gain factor in Eq. (33) and since we want to maximise the gain, we use its inverse as a loss function to minimise. This results in

$$\frac{1}{f_{\text{eff}}} = \epsilon_{\text{baseline}} \left( t_1 \cdot \frac{1}{\epsilon_{\text{1st,surr}} \epsilon_{\text{2nd,surr}}} + t_2 \cdot \frac{1}{\epsilon_{\text{2nd,surr}}} \right) \,, \tag{34}$$

where we absorbed all timing constants into two coefficients $t_1$ and $t_2$. To make the gain factor differentiable, we use the approximations of Eqs. (20),(21) for the surrogate

unweighting efficiencies $\epsilon_{1\text{st,surr}}$ and $\epsilon_{2\text{nd,surr}}$, leading to the inverse of the approximated effective gain factor $\tilde{f}_{\text{eff}}^{-1}$:

$$
\begin{aligned}
\frac{1}{f_{\text{eff}}} &\approx \epsilon_{\text{baseline}}\left(t_1 \cdot \frac{s_{\max}x_{\max}}{\langle s\rangle\langle x\rangle_{\text{weight=s}}} + t_2 \cdot \frac{x_{\max}}{\langle x\rangle_{\text{weight=s}}}\right), \\
&\approx \frac{\epsilon_{\text{baseline}}}{\langle w\rangle}x_{\max}\big(t_1 \cdot s_{\max} + t_2 \cdot \langle s\rangle\big) \equiv \tilde{f}_{\text{eff}}^{-1}.
\end{aligned}
\tag{35}
$$

In the last step, we used $\langle x\rangle_{\text{weight=s}} \approx \langle w\rangle/\langle s\rangle$, assuming $\max(1, \frac{s}{s_{\max}}) \approx 1$. Minimising $\tilde{f}_{\text{eff}}^{-1}$ in this form is not stable, since the normalisation of $s$ is unbounded. Therefore, a regularisation is needed. As a simple fix, we set $s_{\max} = w_{\max}$, rendering this factor independent of the trainable model parameters, collectively referred to as $\theta$. Ignoring the constant prefactor, which is irrelevant for the optimisation, this results in

$$
L_{\text{gf}}(\theta) = x_{\max}(\theta)\big(t_1 \cdot w_{\max} + t_2 \cdot \langle s\rangle(\theta)\big).
\tag{36}
$$

We observe that using this loss function for training directly from the start, a randomly initialised network achieves lower gain factors than using MSE loss. Therefore, we only optimise $L_{\text{gf}}(\theta)$ in a second step, using networks that have been pre-trained with the MSE loss. This approach results in a much more favourable surrogate distribution that leads to significantly higher gain factors, see the discussion in Sec. 3.5. For $\epsilon_x$, defined in Eq. (17), we use a value of 0.1% during training to be compatible with $\epsilon_{\max} = 0.001$ used in the baseline approach.

## 3.3    Hyperparameter optimisation

To fix the architecture of our neural networks and their training, we performed a (limited) hyperparameter optimisation using the OPTIMA [51] framework. To this end, we used one of the most expensive partonic channels, namely $gd \to e^+e^- gggggd$. As a starting point, we used the parameter set found to perform best in a previous study [32] and carried out a series of one- and two-dimensional scans around that. In particular, we varied the number of hidden layers between $[1, 2, 3, 4, 5, 6]$, the number of nodes per layer between $[64, 96, 128, 256]$, and the batch size during training within $[512, 1024, 2048, 8192]$. Furthermore, for the weight initialisation we considered "He uniform" [52] and "Glorot uniform" [53].

For the number of hidden layers we did not observe significant improvements beyond 4 layers. Similarly, for the number of nodes, the performance saturates already at 64 nodes. However, we decided to use 128 nodes in our final setup. For training on the CPU, a batch size of 1024 was found to offer the best performance, with 2048 achieving a similar result. The weight initialiser "He uniform" was found to perform slightly better than "Glorot uniform" and is thus used in the final setup. We furthermore performed a two-dimensional scan of "hidden layers" versus "nodes in hidden layers", but no significantly better results have been achieved. Concerning the change of the loss function from MSE to $L_{\text{gf}}$, this is done after 50 training epochs. Within the tested range of 30 to 60 no significant differences were observed.

The final collection of hyperparameters used in this work is collated in Tab. 4. These are used for all partonic subprocesses for which we train surrogates, independent of parton flavours and final-state multiplicity.

## 3.4    Training

The network training procedure is largely equivalent to the description in [32]. The training data is obtained from the integration phase of SHERPA, where weighted events are gener-

| Parameter | Value |
|---|---|
| Hidden layers | 4 |
| Nodes per hidden layer | 128 |
| Activation function | swish [54, 55] |
| Weight initialiser | He uniform [52] |
| Loss function | 1) MSE on arcsinh 2) gain factor |
| Loss change epoch | 50 |
| Batch size | 1024 |
| Optimiser | ADAM [56] |
| Initial learning rate | $10^{-3}$ |
| Callbacks | EarlyStopping |

Table 4: Neural-network hyperparameters used for the surrogates of partonic subprocesses.

ated to determine the total cross section as well as an estimate of the weight maximum $w_{\mathrm{max}}$, cf. Sec. 2.5. For each partonic subprocess, we write the external particle momenta, matrix-element value and the phase-space weight of 1.5 million events to a file. We partition the dataset into 640 000 events for training, 160 000 for validation during training, and the remaining 700 000 events for subsequent performance evaluations of the model. Training of the dipole model, as described in Ref. [32], employs the KERAS [57] library alongside the TENSORFLOW [58] backend, with models ultimately stored in the ONNX format [59]. To determine $x_{\mathrm{max}}$, the test set (1 million events) is used in conjunction with the maximum-reduction method, setting $\epsilon_x = 0.1\,\%$.

Once the training of the surrogate model has been completed, it is used for event generation with SHERPA. At the point where matrix elements would ordinarily be evaluated by COMIX, the particle momenta of the current trial event are instead used to compute additional features, in particular the phase-space mapping variables $y_{ijk}$, and the kinematic invariants $s_{ij}$. Combined with the momenta, these variables are passed as input to the surrogate model, which is executed in a single CPU thread using the ONNX Runtime C++ API [60]. The output of the model consists of the dipole coefficients $C_{ijk}$, which are then combined with the dipole functions $D_{ijk}$ according to Eq. (30), resulting in

$$s_{\mathrm{ME}} = \sum_{\{ijk\}} C_{ijk} D_{ijk} \,, \qquad (37)$$

with $\{ijk\}$ the three-parton ensemble involved in the dipole function. This yields an approximation $s_{\mathrm{ME}}$ for the true matrix-element weight $w_{\mathrm{ME}}$, which is then used to decide whether the trial event is accepted or rejected, based on the two-step unweighting algorithm described in Sec. 2.

In Figure 2, we show an exemplary loss curve for the process $gu \to e^+e^- ggggu$. Before changing the loss function at epoch 50 from MSE to $L_{\mathrm{gf}}$ we achieve an (approximated) gain factor of $\tilde{f}_{\mathrm{eff}} = 20$, which is improved to $\tilde{f}_{\mathrm{eff}} = 200$ in epoch 68. As there is no further improvement at later epochs, the training is terminated at epoch 128. Note that this significant step is artificially pronounced by setting $s_{\mathrm{max}} = w_{\mathrm{max}}$ in the approximated gain factor. This assumption is not correct for our optimal model because we optimise $\epsilon_s$ according to the gain factor (resulting in $\epsilon_s = 0.018$ for $gu \to e^+e^- ggggu$). The other approximations also modify the overall normalisation of the gain factor and the actually achieved gain factor is only slightly improved with the new loss function and is found at about $f_{\mathrm{eff}} = 70$. However, the new loss function will have a large impact for event

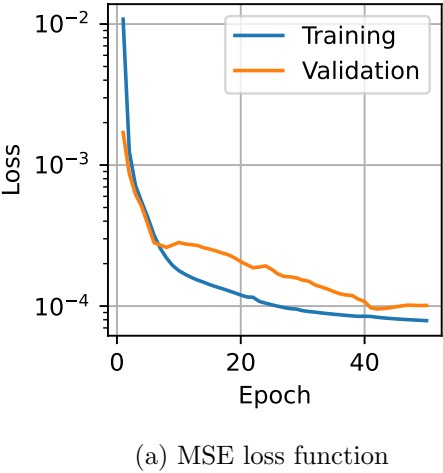
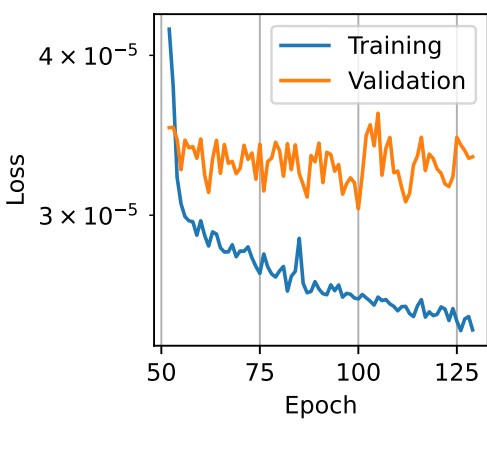

(a) MSE loss function                              (b) Gain loss function

Figure 2: Loss function curves for the MSE and gain-factor training for $gu \to e^+e^- ggggu$.

generation with low $\epsilon_x$, which is briefly discussed in Appendix B.

## 3.5 Evaluation

To investigate the effect of changing the loss function from MSE to $L_{\text{gf}}$, a comparison of their respective results is provided in Figure 3. The distribution of the ratio of the true matrix-element weight over the surrogate, i.e. $\frac{w_{\text{ME}}}{s_{\text{ME}}} \approx x$, in comparison to the true weight $w_{\text{ME}}$, is shown for the process $gu \to e^+e^- ggggu$ after training for 50 epochs using the MSE loss[6] in Fig. 3a. In Fig. 3b the same distribution is shown after training with $L_{\text{gf}}$ for another 18 epochs, when the best approximated validation gain factor is reached. Although there is no significant change in the width of the distribution, a pronounced change can be seen in its shape. The most striking change is that the distribution is no longer symmetric, but skewed towards small values of $x$. There are fewer events with $x$ above the weighted mean $\langle x \rangle_{\text{weight}=\text{s}}$, indicated by the red dashed lines, and more events with $x$ below the mean than for the MSE loss training. Furthermore, a positive correlation can be seen between $x$ and $w_{\text{ME}}$. The change in overall normalisation of $x$ can be ignored since it does not influence the gain factor that depends on $\langle x \rangle_{\text{weight}=\text{s}}/x_{\text{max}}$, cf. Eq. (21).

We remind ourselves that the key towards achieving a large effective gain factor $f_{\text{eff}}$ is a small value of $x_{\text{max}}$, so that the second unweighting efficiency is large. Accordingly, severe underestimation of the true weight $w$ by the surrogate $s$ should be avoided. Overestimation, on the other hand, is less problematic as long as it does not shift the mean $\langle x \rangle_{\text{weight}=\text{s}}$ too far from the maximum $x_{\text{max}}$. Events with low values of the true weight $w_{\text{ME}}$ are generated infrequently, as can be inferred from the marginal distribution of Fig. 3a. Fig. 3b indicates that the gain factor loss function lets the model overestimate the weights of events with low $w$. These events are therefore accepted too often in the first unweighting step, which is worthwhile in order to avoid an increased $x_{\text{max}}$ that would reduce the acceptance rate for all generated points. This results in a systematic overestimation for events with low $w$. The maximum $x_{\text{max}}$ is therefore found where $w$ is large, corresponding to the right hand side of Fig. 3b. The second term of Eq. (36) is insensitive to the normalisation of $x$ since $\langle s_{\text{ME}} \rangle \cdot x_{\text{max}} = \langle w_{\text{ME}}/x \rangle \cdot x_{\text{max}}$. A scaling of $x$ is directly canceled in the ratio by the corresponding scaling of $x_{\text{max}}$. This explains the shift of $\langle x \rangle_{\text{weight}=\text{s}}$ in Fig. 3b.

---

[6]To ease the discussion, we here assume $\epsilon_s \approx 0$, resulting in $\frac{w_{\text{ME}}}{s_{\text{ME}}} \approx x$ and $\frac{\langle w_{\text{ME}} \rangle}{\langle s_{\text{ME}} \rangle} \approx \langle x \rangle_{\text{weight}=\text{s}}$.

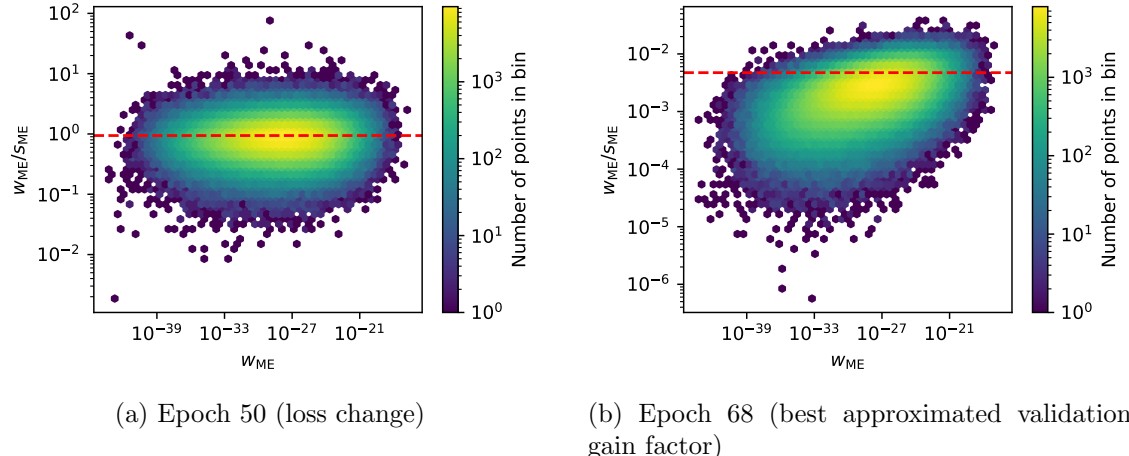

(a) Epoch 50 (loss change)

(b) Epoch 68 (best approximated validation gain factor)

Figure 3: Accuracy $x$ of the surrogate-model prediction in dependence on the weight $w_{\mathrm{ME}}$ for the channel $gu \to e^+ e^- ggggu$. The ratio $\frac{\langle w_{\mathrm{ME}}\rangle}{\langle s_{\mathrm{ME}}\rangle}$ that determines the second-unweighting efficiency is shown as dashed red line.

# 4 Real-life application: $Z$+jets production at the LHC

In the previous section, we have introduced performance measures and demonstrated improvements for individual processes when using surrogate unweighting. In this section we now move on and apply the method for the first time in a fully realistic LHC event-generation scenario. The bulk of computing time for simulation in LHC experiments is spent on a few major samples like vector-boson production ($W$+jets, $Z/\gamma^*$+jets) and top-quark pair production that constitute interesting signals in their own right but, most importantly, contribute as backgrounds in almost all analyses. For an assessment of our method, we therefore here consider the simulation of $pp \to Z/\gamma^*(\to e^+ e^-)$+jets production at the LHC at $\sqrt{s} = 13$ TeV using SHERPA, thereby closely following the default generator setups used by the ATLAS collaboration [45, 61]. We merge tree-level matrix elements with up to six final-state QCD partons provided by COMIX [8] (both for colour summation and colour sampling), matched with SHERPA's Catani–Seymour dipole shower [62] using the truncated-shower scheme [42]. The merging scale is set to $Q_{\mathrm{cut}} = 20$ GeV. The enhancement function introduced in Eq. (10) is used consistently. All settings can be found in the corresponding SHERPA run card in Appendix A.

## 4.1 Realistic time measures

The two-stage unweighting algorithm based on fast neural-network surrogates offers great potential to accelerate the production of unweighted events in phenomenological and experimental applications. However, the actual gain of using a surrogate for a specific process crucially depends on the evaluation times both for the full matrix element and the corresponding surrogate, as well as the efficiencies of the two unweighting steps. Furthermore, due to the occurrence of overweights, the emerging event samples could potentially be statistically diluted, which would require considering larger samples compared to the baseline approach. We therefore need to develop realistic time measures that reflect these competing aspects.

Below, for each of the two approaches, baseline and surrogate unweighting, we construct and measure the relevant total timing for the event generation of a realistic LHC sample from the individual times per unweighted event in two steps: First, we include

the statistical dilution and the remaining part of event generation in an effective time per event, $t_{\text{eff}}^{[\text{approach}]}$. Second, this is integrated over all contributing processes and generated numbers of events according to their cross sections, to finally obtain the total time spent on the multijet-merged simulation with the respective approach, $T_{\text{merged process}}^{[\text{approach}]}$.

### 4.1.1 Time per effective event

The low-level basis for timing measurements are the performance measures $t_{\text{unw}}^{\text{baseline}}$ and $t_{\text{unw}}^{\text{surr}}$ introduced in Sec. 3.1. To make all CPU timings comparable across architectures, they are converted into the HEPScore23 metric (HS23) [63], which yields for our cluster (using AMD EPYC 7702 CPUs) a factor of $1\,\text{s} \approx 20\,\text{HS23s}$. In Tab. 5 we summarise all relevant measures for the two SHERPA approaches, based on colour-summed (Baseline$_\Sigma$) and colour-sampled (Baseline) matrix elements. In particular, we quote average timings for the evaluation of the matrix-element and phase-space weights in HS23 milliseconds. We thereby summarise all results as cross-section weighted averages over all individual subprocesses for each final-state jet multiplicity. Furthermore, we quote the cut and unweighting efficiencies for the baseline approaches. This then results in the timing measure to generate an unweighted event in the baseline approaches, namely $t_{\text{unw}}^{[\text{approach}]}$, quoted in HS23 seconds.

To derive an estimate for the surrogate approach, namely $t_{\text{unw}}^{\text{est. surr}}$, without having to actually train networks for all contributing partonic channels, we assume a universal surrogate evaluation time of $\langle t_{\text{surr}} \rangle = 6\,\text{mHS23s}$, which is slightly faster than the actual evaluation time for $Z + 5$ jets, a first unweighting efficiency of $\epsilon_{\text{1st,surr}} = \epsilon_{\text{baseline}}$ and an estimate for the second unweighting efficiency of $\epsilon_{\text{2nd,surr}} = 0.3$. Based on these assumptions, we estimate for $Z + 6$ jets final states a time per unweighted event with surrogates of $t_{\text{unw}}^{\text{est. surr}} = 7\,744\,\text{HS23s}$, comparing to $t_{\text{unw}}^{\text{baseline}} = 63\,561\,\text{HS23s}$ for the colour-summed baseline and $23\,859\,\text{HS23s}$ for the colour-sampled approach. For $Z + 5$ jets with surrogates, the time per unweighted event is estimated to $t_{\text{unw}}^{\text{est. surr}} = 1\,618\,\text{HS23s}$, instead of $t_{\text{unw}}^{\text{baseline}} = 2\,333\,\text{HS23s}$ for the colour-summed and $7\,268\,\text{HS23s}$ for the colour-sampled baseline. However, for final states with less than 5 jets the colour-summed configuration without surrogates is, in fact, expected to be faster than the surrogate approach. Accordingly, in what follows we will only pursue the use of surrogates for (selected) $Z + 5, 6$-jets processes.

The computing time per effective event has to include two additional contributions: The rest of the event generation chain, which is independent of the approach used for unweighting, is here estimated as $t_{\text{overhead}} \approx 25\,\text{HS23s}$[7], and the statistical dilution due to remaining event-weight spread is accounted for by the Kish effective sample size [64], resulting in

$$t_{\text{eff}}^{[\text{approach}]} \equiv \frac{1}{\alpha^{[\text{approach}]}} \left( t_{\text{unw}}^{[\text{approach}]} + t_{\text{overhead}} \right) . \tag{38}$$

Therein, $\alpha$ is the statistical dilution given by [8]

$$\alpha(\mathbf{p}, \tilde{\mathbf{w}}) \equiv \frac{(\sum_i p_i \tilde{w}_i)^2}{\sum_j p_j \tilde{w}_j^2 \sum_k p_k} = \frac{\langle \mathbf{p}\tilde{\mathbf{w}} \rangle^2}{\langle \mathbf{p}\tilde{\mathbf{w}}^2 \rangle \langle \mathbf{p} \rangle} , \tag{39}$$

---

[7]Note that with $t_{\text{overhead}}$ we here account in particular for the parton-shower evolution and the hadronisation of partons, however, we do not take into account a potential detector simulation. This component is rather experiment specific and might in some cases be up to around 2000 HS23s.

[8]If one would not fix $\epsilon_{\text{max}}$ to 0.001, but instead optimise it to potentially large values, then one might also need to take into account the statistical dilution differentially, e.g. in the tail of distributions due to the local accumulation of large weights. A rigorous approach to this will be discussed in [46].

| $Z+n$ jets $n$ | approach | $\langle t_{\text{PS}}\rangle$ [mHS23s] | $\langle t_{\text{ME}}\rangle$ [mHS23s] | $\epsilon_{\text{baseline}}$ | $t_{\text{unw}}^{[\text{approach}]}$ [HS23s] | $t_{\text{unw}}^{\text{est. surr}}$ [HS23s] |
|---|---|---|---|---|---|---|
| 6 | Baseline$_\Sigma$ | 358.7 | 9 659.5 | 0.000 158 | 63 561 | 7 744 |
|   | Baseline | 328.0 | 106.4 | 0.000 018 | 23 859 | |
| 5 | Baseline$_\Sigma$ | 131.1 | 528.6 | 0.000 283 | 2 333 | 1 618 |
|   | Baseline | 122.1 | 33.2 | 0.000 021 | 7 268 | |
| 4 | Baseline$_\Sigma$ | 60.9 | 51.3 | 0.001 27 | 89 | 176 |
|   | Baseline | 60.7 | 11.8 | 0.000 17 | 434 | |
| 3 | Baseline$_\Sigma$ | 32.2 | 9.3 | 0.003 46 | 12 | 37 |
|   | Baseline | 33.3 | 4.7 | 0.000 64 | 60 | |
| 2 | Baseline$_\Sigma$ | 17.3 | 2.8 | 0.009 9 | 2.0 | 7.9 |
|   | Baseline | 17.4 | 2.0 | 0.005 2 | 3.7 | |
| 1 | Baseline$_\Sigma$ | 8.9 | 1.1 | 0.063 | 0.16 | 0.80 |
|   | Baseline | 7.0 | 0.9 | 0.031 | 0.26 | |
| 0 | Baseline$_\Sigma$ | 5.3 | 0.6 | 0.20 | 0.029 | 0.19 |
|   | Baseline | 4.1 | 0.4 | 0.20 | 0.022 | |

Table 5: Average performance measures as defined in Sec. 3.1 for exclusive jet multiplicities in the $Z$+jets setup described in the text. Shown are results based on using colour-summed (Baseline$_\Sigma$) and colour-sampled (Baseline) matrix elements. To estimate the time per unweighted event in the surrogate approach, $t_{\text{unw}}^{\text{est. surr}}$, see Eq. (32), we here assume $\epsilon_{\text{1st,surr}} = \epsilon_{\text{baseline}}$, $\epsilon_{\text{2nd,surr}} = 0.3$, and $\langle t_{\text{surr}}\rangle = 6\,\text{mHS23s}$.

where $p_i$ is the probability to keep an event after (partial) unweighting with weight $\tilde{w}_i$. For baseline unweighting, $\alpha$ is given by

$$\alpha^{\text{baseline}} = \alpha\left(\mathbf{p}_{\text{baseline}}, \mathbf{w}_{\text{overweight}}\right). \tag{40}$$

For the two-step partial unweighting with surrogates it results in

$$\alpha^{\text{surr}} = \alpha\left(\mathbf{p}_{\text{surr}}, \mathbf{x}_{\text{overweight}}\right). \tag{41}$$

### 4.1.2 Total time for event generation

We study an event-generation scenario for the HL-LHC, where we aim for a simulation sample size corresponding to twice the integrated luminosity, i.e. $2 \times \mathcal{L}_{\text{HL-LHC}}$, assuming $\mathcal{L}_{\text{HL-LHC}} = 3000\,\text{fb}^{-1}$. Having more events in Monte Carlo samples than expected in real data is a requirement by most modern analyses, as they rely on machine-learning based discriminators which need to be trained with an abundance of signal and background events. The multiplicator of 2 chosen above is empirically taken from the size of the aforementioned ATLAS $Z$+jets sample. Due to the phase-space enhancement, quantified in Sec. 2.4, this yields an even higher number of effective events in the enhanced jet multiplicities, i.e. $N_{\text{evt,tot}} = 2 \times 3000\,\text{fb}^{-1} \cdot \langle h\rangle \cdot \sigma$.

The total time needed for a given process then consists not only of the time spent on event generation, but also has to include the initial phase-space optimisation and integration phases. Thus, the total time spent on an individual subprocess is:

$$T_{\text{subprocess}}^{[\text{approach}]} \equiv N_{\text{evt,tot}} \cdot \frac{1}{\epsilon_{\text{Sudakov}}} \cdot t_{\text{eff}}^{[\text{approach}]} + (N_{\text{opt}} + N_{\text{int}}) \cdot (\langle t_{\text{ME}}\rangle + \langle t_{\text{PS}}\rangle) \tag{42}$$

with:

$$\epsilon_{\text{Sudakov}} = \text{average efficiency of the Sudakov event veto,}$$
$$N_{\text{opt}} = \text{number of points generated for integrator optimisation,}$$
$$N_{\text{int}} = \text{number of points generated for integration and surrogate NN training.}$$

The Sudakov veto efficiency thereby accounts for the fact that in the truncated-shower merging scheme used here, matrix-element configurations get rejected when during their parton-shower evolution emissions above the merging cut $Q_{\text{cut}}$ appear. For the process studied here, the average efficiency of the Sudakov event veto is $\epsilon_{\text{Sudakov}} = 0.23$. The total number of points generated during the integrator optimisation and the initial integration phase is approximately 5 million for the phase-space integrator settings used here, see App. A. However, for large-scale production campaigns it might be worth to further optimise these choices.

The time per process group, e.g. $Z+$ exactly 6 jets in the matrix element, is then simply the sum of the times spent on all contributing subprocesses, i.e.

$$T_{\text{process}}^{[\text{approach}]} \equiv \sum_{\text{subprocesses}} T_{\text{subprocess}}^{[\text{approach}]} . \tag{43}$$

Consequently, the total time spent on the fully realistic merged setup is then:

$$T_{\text{merged process}}^{[\text{approach}]} \equiv \sum_{\text{processes}} T_{\text{process}}^{[\text{approach}]} . \tag{44}$$

It presents the final figure of merit to decide which approach to use.

## 4.2 Results

With a state-of-the-art LHC event generation setup and realistic timing measures defined, we can now compare the results obtained with the surrogate and baseline approaches. In Table 6 we present results for $T_{\text{merged process}}^{[\text{approach}]}$, measured in HS23 Mega-years, considering the production of samples with an increasing number of matrix-element jets included. As non-surrogate approaches we again consider two options, based on colour-summed ("Baseline$_\Sigma$") and colour-sampled ("Baseline") matrix-element evaluation. As discussed above, colour sampling is the current default in SHERPA and used most heavily by the LHC experiments, but colour-summed matrix elements from COMIX offer a competitive alternative for $Z+ \leq 5$ jets production, where the total sample-generation time can be reduced by a factor $47.83/16.09 \approx 3$. In future versions of SHERPA we will offer the option to pick the colour treatment specifically for each parton multiplicity contributing to a multijet-merged calculation.

Considering the ratio of the surrogate approach to the best baseline approach one finds that the event generation can be sped up significantly, reducing the CPU budget by a factor of 11.1 in the state-of-the-art setup for $Z+ \leq 6$ jet production. If one were to include only up to five jets in the simulation, the improvement would still be a factor of 3.3 with respect to Baseline$_\Sigma$. For samples including matrix elements only for 4 or less partons there is no net gain from unweighting with surrogates and these should thus be generated in the colour-summed approach (Baseline$_\Sigma$). This is a reflection of (and switched automatically by) the individual event-timing estimates quoted in Tab. 5.

Based on the above estimates for the potential of surrogate unweighting, we here restrict the use of surrogates to $Z + 5$ and $Z + 6$ jets processes. From a total of 112 such subprocesses, cf. Tab. 2, we select the 70 with the highest improvement potential

| Process | $T_{\text{merged process}}^{[\text{approach}]}$ [MHS23y] | | | Ratio |
|---|---|---|---|---|
| | Baseline$_\Sigma$ | Baseline | Surrogate | |
| Z + ≤0 jets | 0.04 | 0.04 | - | |
| Z + ≤1 jets | 0.11 | 0.11 | - | |
| Z + ≤2 jets | 0.32 | 0.34 | - | |
| Z + ≤3 jets | 0.67 | 1.23 | - | |
| Z + ≤4 jets | 1.60 | 5.44 | - | |
| Z + ≤5 jets | 16.09 | 47.83 | 4.91 | 3.3 |
| Z + ≤6 jets | 297.73 | 130.51 | 11.78 | 11.1 |

Table 6: Computational time $T_{\text{merged process}}^{[\text{approach}]}$ needed to simulate event counts corresponding to $2 \times \mathcal{L}_{\text{HL-LHC}}$ for the three considered calculational approaches, see text for details. The overall optimal approach is shown in green, while the best baseline approach is marked blue. Their ratio is shown in the last column. Note, absolute computational times for merged setups with less than 6 final-state jets are conservatively estimated, as we use the Sudakov efficiency measured for the 6-jet process of $\epsilon_{\text{Sudakov}} = 0.23$ throughout.

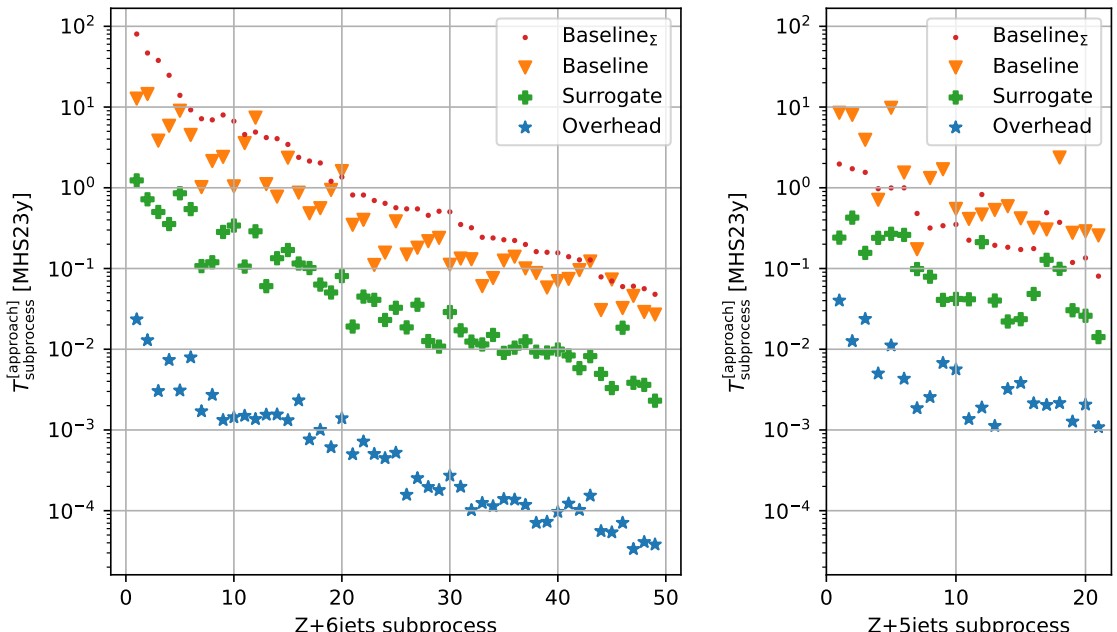

Figure 4: Total compute time per subprocess for $Z + 6$ jets (left) and $Z + 5$ jets (right) channels contributing to inclusive $Z+ \leq 6$ jets production. For the overhead time we here consider $t_{\text{overhead}} = 25 \, \text{HS23s}$ per event. The channels are sorted on the $x$-axis according to their estimated time reduction.

of the absolute computational time with the surrogate approach, and train corresponding neural networks. These represent the top 49 of the $Z + 6$ jets subprocesses, and the top 21 contributing to $Z + 5$ jets production. While using surrogates for the remaining 42 subprocesses could in principle further reduce the total computational time by up to 9%, these reductions are rather insignificant and are thus not considered here. In Fig. 4 we collate the total compute times in the three considered approaches for the sets of $Z + 6$ jets (left panel) and $Z + 5$ jets (right panel) subprocesses for which we trained neural-network surrogates. The quoted numbers are based on measured efficiencies and runtimes of the surrogates. We thereby assume an overhead time per event, corresponding to parton showering, hadronisation of partons, and the generation of the underlying event, of $t_{\text{overhead}} = 25\,\text{HS23s}$ per event. Accordingly, the overhead contribution to the total time of the subprocesses (blue stars) scales with the (enhanced) cross section of the process. It can clearly be seen that for all considered channels the surrogate approach (green crosses) performs best. While for the $Z + 6$ jets channels the colour-sampled baseline (orange triangles) is typically faster than colour summation (red dots), it is the opposite for the $Z + 5$ jets channels. Accordingly, even without using surrogates, for $Z$+jets production the compute time of the SHERPA default can be significantly reduced by using colour summation for processes with less than 5 final-state jets.

In Figs. 5 and 6 we present the relative contributions of the different subprocesses to the total CPU budget in the SHERPA baseline approach and when using surrogates, respectively. While only a few processes dominate the computing time in the baseline approach mainly due to their low unweighting efficiency, these are accelerated most in the surrogate approach and thus the relative contributions are largely levelled out.

The extrapolated time savings can also be checked directly in the actual event generation. We produce samples of 10 000 events, a typical chunk size used by LHC experiments. In Tab. 7 we collate the respective times in HS23 hours for the three considered approaches for the generation of inclusive $Z+ \leq 6$ jets event samples. Besides the total times spent on generating events, we also quote the time needed to evaluate scattering matrix elements. We recognise that the relative contribution of the matrix-element evaluations to the overall compute time shrinks from 96%, when using colour-summed matrix elements, to 23% when using colour sampling, and as little as 15% in the surrogate approach. In the last column we quote the number of events that can be generated per day, when using 20 HS23 cores. These increase from $\mathcal{O}(100)$ for the baseline approaches to $\mathcal{O}(1\,000)$ when using surrogates for the dominant $Z+5$ jets and $Z+6$ jets channels. So the gain from using surrogates is about a factor of 11 in CPU resources for this example. Note that the ratios of total compute times, i.e. events per day, quoted here are not necessarily identical to the ones extrapolated from the sum of all subprocesses given before. In fact, the overhead time, dominated by the shower evolution of the events, depends on the jet multiplicity. However, from the numbers obtained, this correction appears to be rather subleading.

In Fig. 7 we present a breakdown of the total compute time into the different parts of the event generation process when using the surrogate-based approach. We thereby consider the phase-space generation including the evaluation of cuts (PS), the PDF weight evaluation (PDF), the evaluation of the surrogates (Surr) and actual matrix elements (ME), and, lastly, the time spent on parton showering, hadronisation and underlying events (Overhead). As already seen in Tab. 7, surrogate unweighting not only reduces the total compute time, but also the relative contribution of the time needed to evaluate matrix elements is significantly reduced. As a consequence, it becomes apparent that in order to further reduce the total compute time, besides aiming for yet better surrogates, it is worthwhile to revisit the other components of the event-generation process. A similar conclusion holds for the SHERPA baseline approach based on colour-sampled matrix

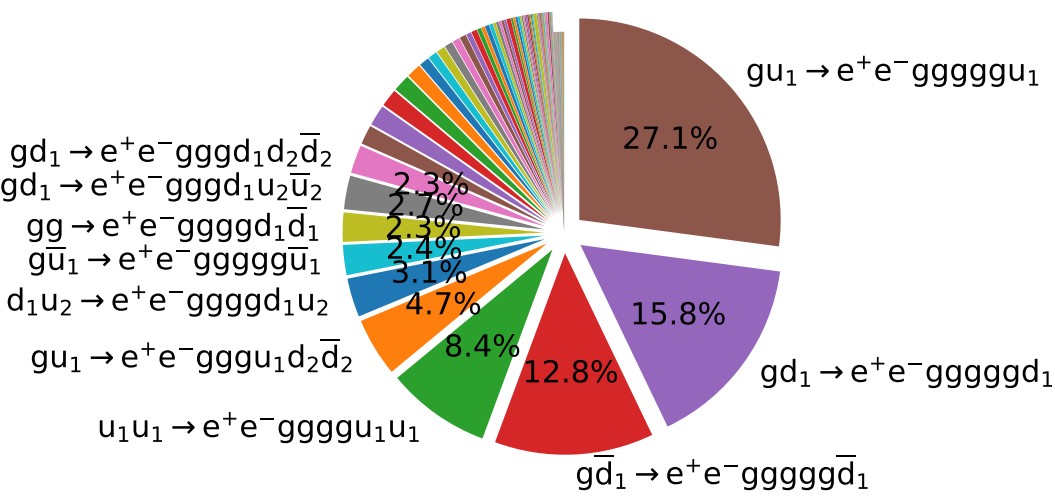

Figure 5: Contribution per subprocess to the total computation time for $\epsilon_{\max} = 0.001$ of inclusive $Z+ \leq 6$-jets with overhead time estimate of 25 HS23s/event in the SHERPA baseline approach, i.e. without surrogates. Labels are shown for processes with more than 2% contribution to the total compute time. Subprocesses for which a surrogate got trained for are moved outwards.

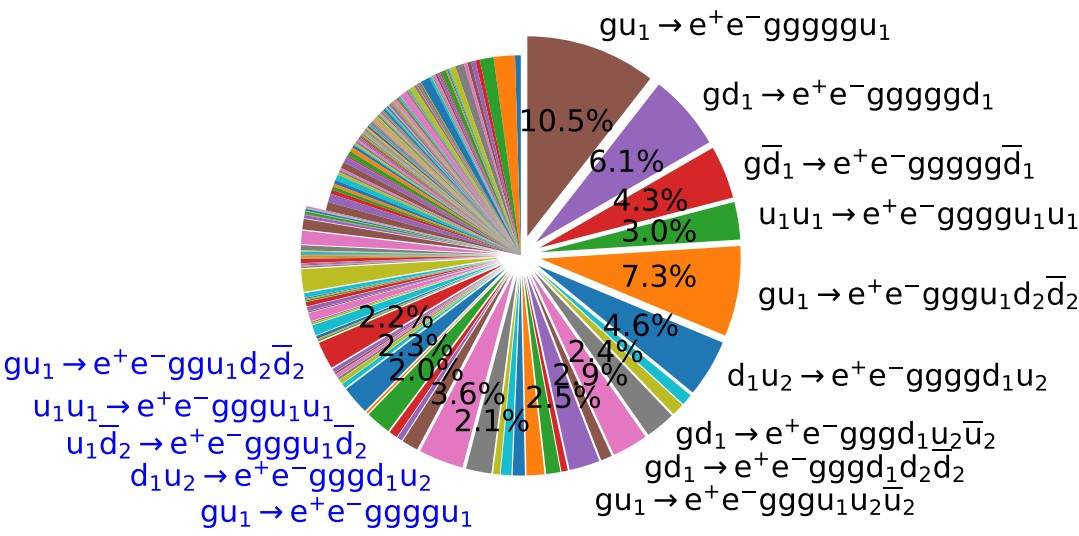

Figure 6: Same as Fig. 5 but for surrogate unweighting. Subprocesses with 6 jets in the final state are shown with black labels, while subprocesses with 5 jets in the final state are labelled in blue.

| Approach | $\sum$ Generation time [HS23h] | $\sum$ ME evaluation time [HS23h] | Events/day [on 20 HS23] | relative gain |
|---|---|---|---|---|
| Baseline$_\Sigma$ | 87 929 | 84 392 | 55 | 0.45 |
| Baseline | 39 511 | 9 059 | 121 | 1.0 |
| Surrogate | 3 400 | 505 | 1 412 | 11.7 |

Table 7: Measured event-generation and matrix-element-evaluation times, as well as the corresponding number of events generated per day using 20 HS23 cores, in the case of inclusive $Z+ \leq 6$ jets production for the different approaches. Based on samples comprising 10 000 partially unweighted events. The last column presents the relative gain for the number of events per day with respect to the colour-sampling baseline.

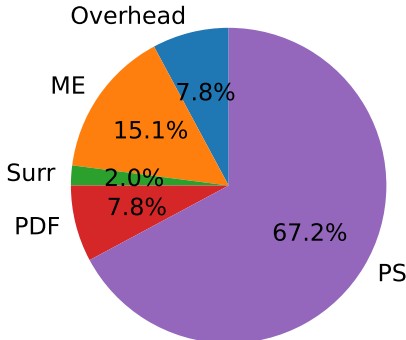

Figure 7: Contribution of the various parts of the event generation to the total computation time for inclusive $Z+ \leq 6$-jets production when using the surrogate approach. The measurements are based on the actual generation of 10,000 partially unweighted events. We thereby considered surrogates for the dominant 70 subprocesses, see text for details.

elements. Most prominently, the evaluation of phase space now dominates. To reduce this component, the efficiency of the phase-space sampler to hit the fiducial phase space, i.e. $\epsilon_{\rm cut}$, should be improved. Furthermore, the actual sampling method, for SHERPA this is a multi-channel method [7,8], and in particular the evaluation of the eventwise phase-space weight contributes significantly. Both aspects can be addressed by novel sampling algorithms, e.g. based on neural importance sampling, as they are currently being developed [17–26]. To speed up the evaluation of the matrix elements, new computational methods employing GPUs are currently being developed for use with SHERPA [15,65].

## 4.3 Validation

The two-step unweighting approach with surrogates should lead to an unbiased event sample, i.e. the physics predictions should be compatible with the baseline event generation within statistical uncertainties. This has been validated thoroughly in previous publications for single processes and here we confirm this agreement also in the case of our realistic multijet-merged $Z$+jets simulations for the LHC.

We use the RIVET framework [66] and its built-in `MC_ZINC` and `MC_ZJETS` analyses to apply physical selection criteria and construct observables for this comparison. Events are selected that contain exactly one electron and one positron, each with $p_{\perp,e^\pm} \geq 25\,\mathrm{GeV}$ and within $|\eta_{e^\pm}| < 3.5$, dressed with prompt photons within $\Delta R < 0.2$ around the lepton and

yielding an invariant mass of $66\,\text{GeV} < m_{e^+e^-} < 116\,\text{GeV}$. Jets are reconstructed from the remaining particles in the event, excluding neutrinos, using the anti-$k_t$ algorithm [67,68] with a radius parameter of $R = 0.4$ and $p_{\perp,j} \geq 20\,\text{GeV}$.

In Fig. 8 we present selected results for observables involving the leptons, i.e. the dilepton invariant mass and transverse momentum, and the final-state jets, namely the exclusive jet multiplicity and the transverse momenta for the leading six jets. For all observables we show the SHERPA baseline result, based on unweighting the exact colour-sampled COMIX matrix elements, and compare to a sample generated with surrogate unweighting, where surrogates are used for selected 5- and 6-parton final states. For the jet-multiplicity and the dilepton-$p_T$ distribution the blue dashed and dotted lines highlight the contributions from $Z+5$ and $Z+6$-parton hard-scattering matrix elements that clearly dominate the tails of the distributions. Furthermore, note that the $p_T$ distribution for the $i$th jet has been rescaled by a factor $10^{1-i}$ for readability. In the lower panels of all plots, the ratios of the surrogate-based predictions and the baseline results are shown. It can be seen that the surrogate-based results are all statistically fully compatible with the baseline SHERPA simulation. This is a confirmation and reminder of the fact that the surrogate approach including a second unweighting provides a reliable, fast, and unbiased prediction.

# 5   Conclusions

In this article, we studied the application of a recently proposed two-staged event-unweighting algorithm based on neural-network surrogates for scattering matrix elements [31] in a realistic multijet-merged simulation of $Z$+jets final states in proton–proton collisions. In previous applications only individual partonic processes have been addressed, guided by the requirement to find a fast and accurate approximation for the corresponding matrix elements by means of neural networks. In Ref. [32] a dedicated network architecture based on the factorisation of QCD matrix elements in the soft- and collinear limits has been considered. For partonic channels contributing at the tree-level to $Z + 4, 5$ jets and $t\bar{t} + 3, 4$ jets production, a reduction in the computational costs for unweighting events between 16 and 350 have been achieved.

The generalisation to multijet-merged simulations requires measures to decide for which contributing partonic processes the usage of surrogates leads to a reduction in computing time. Furthermore, the effect of the parton shower by means of Sudakov vetoes, the mapping of partonic processes that differ by their PDF weights only, or the deliberately biased sampling for rare event kinematics need to be accounted for. In this work we have addressed these generalisations and derived realistic timing estimates for the generation of an effective event, that also accounts for the time spent in event phases after the hard process, as well as the statistical dilution due to the appearance of overweights. The results obtained for surrogate unweighting have been compared to two baseline approaches available in SHERPA-3.0, based on colour-summed and colour-sampled matrix elements from the COMIX generator. We found that up to multiplicities of $Z + 5$ jets colour summation outperforms colour sampling. However, for $Z + 5$ jets and $Z + 6$ jets processes the best results are obtained with surrogate unweighting. To this end, we identified the top 49 $Z+6$ jets and 21 $Z+5$ jets partonic subchannels, with respect to their runtime contribution, and trained corresponding neural-network surrogates. When training the networks, after an initial training phase based on mean squared error loss, we switch to an approximate form of the gain factor as the adaptation incentive. Accordingly, the networks are ultimately optimised to reduce the overall computational cost, rather than simply providing a close approximation of the true event weight throughout the available phase space.

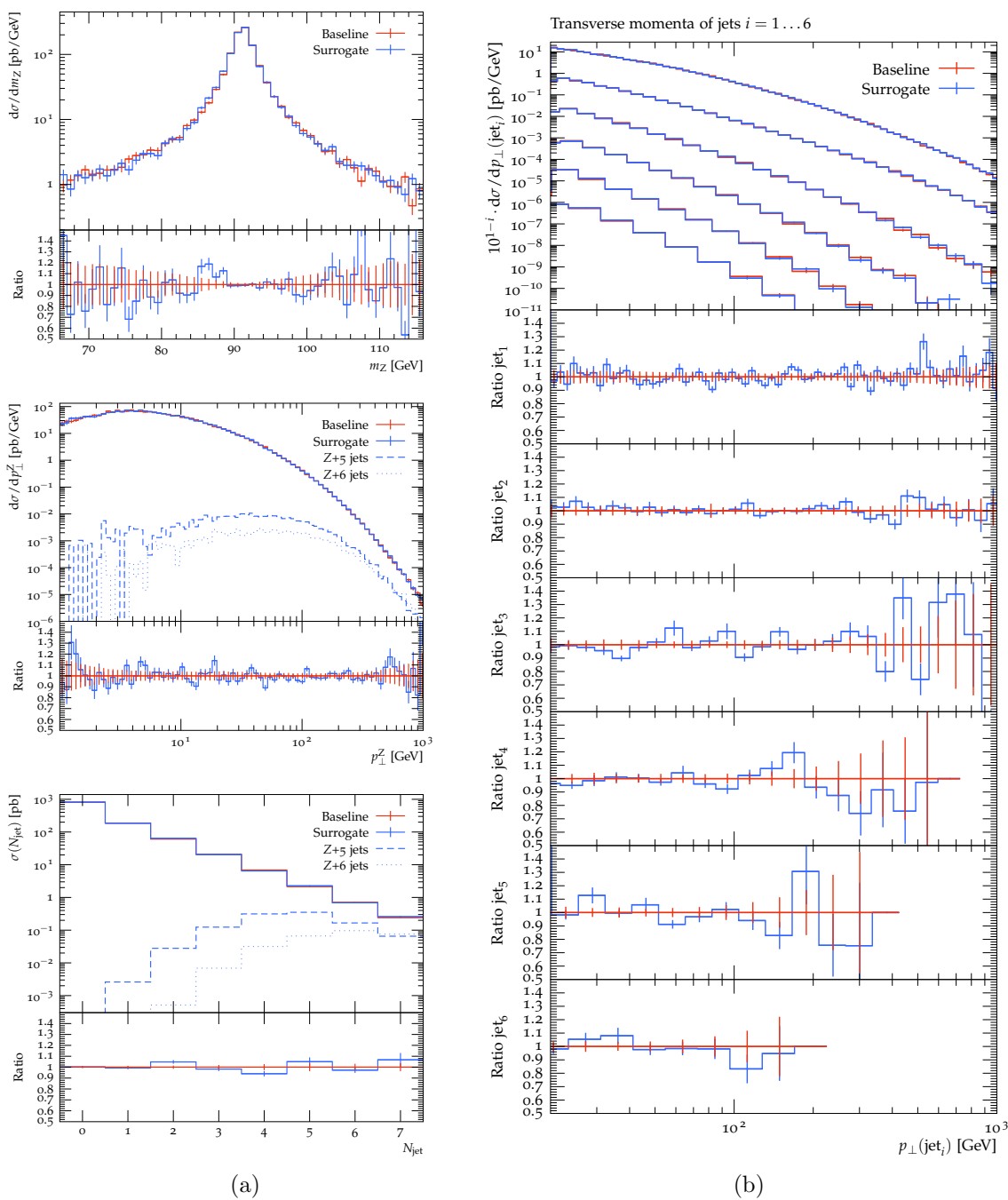

Figure 8: Validation of surrogate predictions against results from a baseline SHERPA simulation in (a) the invariant mass and transverse momentum of the lepton pair and the exclusive multiplicity of jets, and in (b) the transverse momenta of the 6 leading jets. The dashed and dotted sub-contributions in some of the plots denote the 5- and 6-parton processes, in which surrogates are used.

For the three approaches considered, we estimate the computational time needed to generate inclusive $Z$+jets event samples corresponding to twice the projected total integrated luminosity of the HL-LHC, i.e. $2 \times 3\,\mathrm{ab}^{-1}$. For samples including matrix elements with up to 5 final-state partons, surrogate unweighting reduces the overall computational costs by a factor of 3.3, with respect to the fastest baseline approach (based on colour-summed matrix elements). When further including 6-jet matrix elements, surrogate unweighting performs about 11 times faster than the fastest baseline method (based on colour-sampled matrix elements). The estimated computational costs can be reduced from 131 MHS23y to just 12 MHS23y, i.e. about 100 HS23 Mega-years compute time can be saved without any compromise on the physics predictions.

Consequently, the generation of full-statistics samples of $Z$+jets events for HL-LHC data analyses, including matrix elements with up to 6 final-state partons, is clearly feasible. However, it remains to be studied, by how much the resource needs for the next dominant event sample, namely top-quark pair production, can be reduced through surrogate unweighting. Furthermore, the application of surrogates for one-loop-accurate matrix elements should be considered in multijet-merged calculations. First steps towards the regression of one-loop matrix elements have recently been reported [36, 38, 39, 41].

# Acknowledgements

We acknowledge financial support from the German Federal Ministry of Education and Research (BMBF) in the ErUM-Data action plan through the KISS consortium (Verbundprojekt 05D2022). SS is also grateful for support from the German Research Foundation (DFG) through project 456104544 and from BMBF through project 05H24MGA. TJ expresses his gratitude to CIDAS for supporting him with a fellowship.

# A  Sherpa run card settings

In this appendix we provide the settings used for the event-generation runs with SHERPA-3.0 [35].

```
EVENT_GENERATION_MODE: PartiallyUnweighted

# discard hadronisation and underlying event
MI_HANDLER: None
FRAGMENTATION: None

# disable QED corrections
ME_QED: {ENABLED: false}
YFS: {MODE: None}

# collider setup 13 TeV LHC
BEAMS: 2212
BEAM_ENERGIES: 6500

# MEs from Comix, color summed vs sampled
ME_GENERATORS: Comix
COLOR_SCHEME: 1 # 1 summed # 2 sampled (default)

# phase space integrator settings
PSI: {MAXOPT: 2, NOPT: 5, ITMIN: 200000., NPOWER: 0.6}

# process declaration
PROCESSES:
- 93 93 -> 11 -11 93{6}:
    Order: {QCD: Any, EW: 2}
    CKKW: 20
    Max_Epsilon: 1e-3
    2->3-8:
        Enhance_Function: VAR{max(pow(sqrt(H_T2)-PPerp(p[2])-PPerp(p[3]),2)
            ↪ ,PPerp2(p[2]+p[3]))/400.0}
        Max_N_Quarks: 4

# phase space cuts
SELECTORS:
- [Mass, 11, -11, 66, E_CMS]

# event analysis settings
ANALYSIS: Rivet
RIVET:
  --analyses:
    - MC_ZJETS
    - MC_ZINC
  --ignore-beams: 1
  JETCONTS: 1
```

# B    Actual gain factor during training

We here considered training the surrogate models initially with MSE loss and changing after 50 epochs to $L_{\mathrm{gf}}(\theta)$ as the loss function, which aims to maximise the *approximated* gain factor $\tilde{f}_{\mathrm{eff}} \propto 1/L_{\mathrm{gf}}$, see Sec. 3.2. For comparison, in Figs. 9 and 10 we show the *actual* gain factor $f_{\mathrm{eff}}$ for the dominant subprocess $gu \to e^+ e^- ggggg u$, for different values of $\epsilon_s$ during training. While in Fig. 9 we assume $\epsilon_{\max} = \epsilon_x = 0.001$, Fig. 10 corresponds to the case of $\epsilon_{\max} = \epsilon_x = 10^{-15}$. The red line, with $\epsilon_s = 0.02$, in Fig. 9 shows the qualitative behaviour of the approximated gain factor, described in Sec. 3.4. In practice one can freely pick the $\epsilon_s$ value used for event generation. In each single epoch, one would pick the $\epsilon_s$ which maximises the actual gain factor. This means that only the upper envelope of all curves in Fig. 9 is of interest. When looking at the upper envelope, there is a slight improvement in the actual gain factor after the loss change in epoch 50.

Moreover, the increase in gain factor is larger when unweighting against a larger weight maximum, such as when using $\epsilon_x = 10^{-15}$, see Fig. 10. The reason for the larger increase in gain factor is the existence of fewer large overweights from the first unweighting step, which is described in Sec. 3.5. The large overweights from the first unweighting step translate into large $x$ values, increasing $x_{\max}$, which in turn reduces the efficiency of the second unweighting step, which results finally in a worse gain factor. The effect of those large overweights is only very slightly visible in Fig 9, because already $\epsilon_{\max} = \epsilon_x = 0.001$ is large enough to be rather insensitive to those overweights. Note that the absolute value of the gain factor after the loss change is roughly the same in both cases, Figs. 9 and 10.

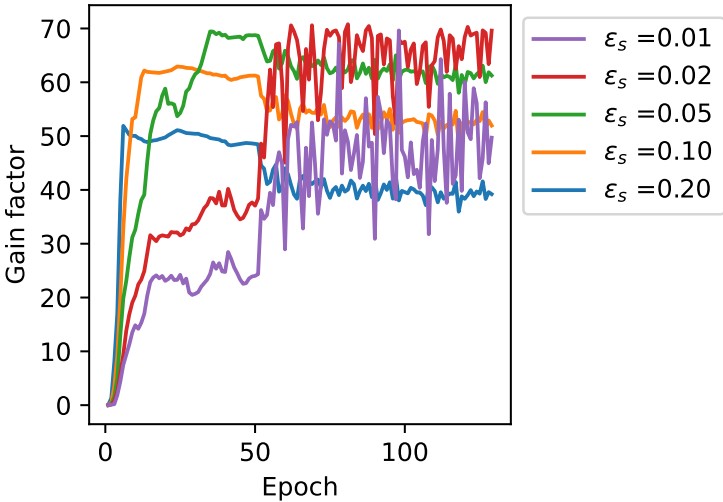

Figure 9: Actual gain factor $f_{\mathrm{eff}}$ during training for the $gu \to e^+ e^- ggggg u$ channel, assuming different $\epsilon_s$ values, thereby assuming $\epsilon_{\max} = \epsilon_x = 0.001$.

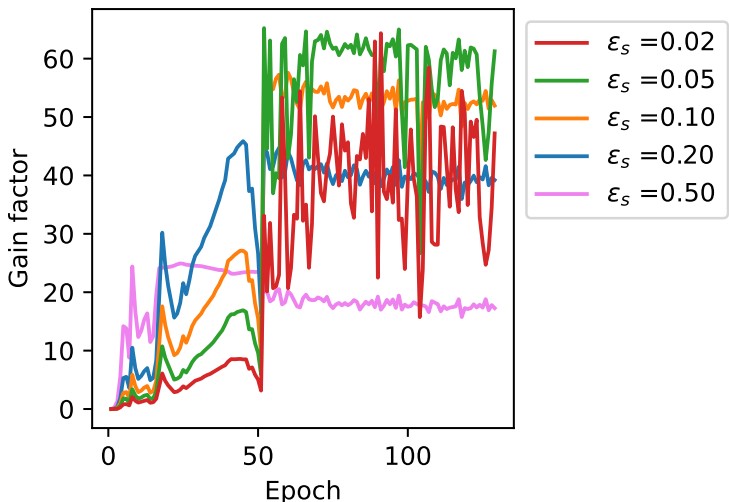

Figure 10: Actual gain factor $f_{\text{eff}}$ during training for the $gu \rightarrow e^+ e^- ggggu$ channel, assuming different $\epsilon_s$ values, thereby assuming $\epsilon_{\max} = \epsilon_x = 10^{-15}$. Note that here the value of $\epsilon_{\max}$ only refers to the event generation, while the value of $w_{\max}$ used in Eq. (36) for the training is the same as in Fig. 9, i.e. the same surrogate is used.

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
