# Peer review of "Accelerating multijet-merged event generation with neural network matrix element surrogates"

_SciPost Physics_

## Round 2 · Referee Report · Anonymous (Referee 1) · 2025-7-29

Strengths

1- The authors make the case and then show very effectively the impact of surrogate models (and colour-sampling vs colour-summing) on timing requirements for event generation. 2- The authors incorporate many subtleties of event generators that are vital for realistic implementations. 3- The authors validate this in a realistic example. 4- The algorithm is detailed, and the plots are useful.

Weaknesses

1- Due to the many aspects of the study, the text may benefit from some additional clarifications and connections between sections. 2- The derivation of the efficiencies and the explanation of the modified $x$ are slightly unclear (see below).

Report

In this work, the authors showcase the utility of surrogate neural networks in the context of event generation. In particular, the authors make a very compelling case that surrogate models would greatly alleviate the computational burden needed to match the needed simulation statistics for the HL-LHC. To do so, they extend previous work on partial unweighting, improving the efficiency of the second step by modifying the bookkeeping of the weights, and include practical necessities such as parton shower matching, biased phase space sampling and multiprocess sampling. To showcase the utility of surrogate modelling, they include a Z+jets benchmark where they showcase how surrogate modelling impacts the highest-multiplicity without sacrificing exactness.

The manuscript is detailed and compelling and the subject matter is of high relevance. I only have a few suggestions and questions, detailed in the requested changes.

Requested changes

Requested changes:

Minor changes / questions:

1- The conditioning in Eq. 21 is not clear. The weight min($s$,$s_{max}$) in Eq. 19 originates from the probability of belonging to the accepted subset (where overweights are always accepted and thus saturate to 1) but Eq. 21 seems to consider only the non-overweights which have weight $s$, and not the overweights. I assumed this is because for small enough $\epsilon$, overweights need not be considered (same goes for Eq. 20 and as indicated in footnote 3). Could the authors clarify whether this is the case? 2- Since the modification introduced in Eq. 16 is shown to be effective in enhancing the performance, it would benefit from a more detailed explanation as to why it works. The figure and the related discussion already show that this is because overweights are kept more or less constant while the second step efficiency increases, but the authors could clarify why this is the case specially because the notation in the cited master thesis is slightly different (and $x$ is not re-defined, just the weight at least in the version found here https://amslaurea.unibo.it/id/eprint/28767/1/Improving_MadGraph5.pdf). 3. Additionally, how is $s$ computed for Fig. 1? Is it with the same surrogate model or with two surrogate models trained with different definitions of $x$? 4- p.16 Eq. 34. Here, $\langle t_{surr}\rangle$ is assumed to be fixed and consequently not optimized. For a fixed architecture, this is clearly correct. However, there is an optimization of architecture described in 3.3. How does the change in architecture impact $\langle t_{surr}\rangle$? I assume it is a minor effect, but the authors should comment on this. 5- p.16 Below Eq. 35, $\langle x\rangle_{s}\approx \frac{\langle w \rangle}{\langle s \rangle}$ assumes $w$ and $s$ are independent. While I understand this is an empirical assumption meant to facilitate the optimization of $s$ with a suitable loss function, $w$ and $s$ should be correlated for a good surrogate model. Are there any alternatives to this assumption as for example using $\langle x \rangle (\theta)$ or is it too unstable? 6- p.17 Below Eq. 36, the authors observe that a randomly initialised network achieves lower gain factors than using MSE loss. Could the authors expand on this? Is this some pathology related to the fact that, for gain purposes, the overall scale of x is not important?

Very minor changes / questions:

1- p. 3 “unit-weight events [...] are needed” This is explained in Sec 2.5, but perhaps a simple “unit-weight events [...] are preferred to avoid statistical dilution due to increased variance” (if this is what the authors mean) would increase clarity. 2- p. 8 When introducing biased phase space sampling, I would add below Eq. 9 that even if certain phase-space configurations are enhanced, the final sample is unbiased, and reference Eq. 27. 3- Ref. 47 is cited as the original source of Eq. 16. Perhaps the authors should include a link if possible. 4- The end of 2.5 justifies Eq. 16 by showing how the fraction of overweights can be kept constant while ensuring higher efficiencies for the second step. This becomes important when exploring the gain factor in Eq. 33, maybe it would be better to highlight this with a brief reference to said equation at the end? 5- Training time is not included in the gain factor, even if it’s included in the overall timing evaluation in section 4. I assumed this is because the training time itself, whatever it ends up being, will be very minor compared to generation time (similarly to the pre-running used to optimize integration). Is this a correct interpretation? 6- The gain factor loss seems pretty unstable even after regularization, especially in validation. Could the authors comment on this and whether it’s an issue for training? The authors could also highlight that suboptimal training is not necessarily a problem as long as the gain factor is large enough, since unbiased results are always achieved. 7- p.22 section 4.1.2, “ Having more events in Monte Carlo samples than expected in real data is a requirement by most modern analyses, as they rely on machine-learning based discriminators which need to be trained with an abundance of signal and background events.” This is true also for non ML-based analysis using “traditional” variables, and it is simply to reduce the MC-stats systematics in any MC-based analysis. I highlight this because this renders the increase in efficiency due to surrogate modelling even more important. 8- p.30 Regarding the last sentence, could the authors comment on the impact, if any, of negative weights on partial unweighting and the use of surrogate models when going beyond LO computations? Is the correction from Eq. 3 to Eq. 16 equally applicable to negative event weights as detailed in Ref. 31?

Recommendation

Ask for minor revision

  • validity: top
  • significance: top
  • originality: high
  • clarity: high
  • formatting: perfect
  • grammar: excellent

Author:  Tim Herrmann  on 2025-09-22  [id 5843]

(in reply to Report 1 on 2025-07-29)

Warnings issued while processing user-supplied markup:

  • Inconsistency: Markdown and reStructuredText syntaxes are mixed. Markdown will be used.
    Add "#coerce:reST" or "#coerce:plain" as the first line of your text to force reStructuredText or no markup.
    You may also contact the helpdesk if the formatting is incorrect and you are unable to edit your text.

We thank the referee for the thorough reading of our manuscript and the very helpful comments and suggestions. We had been waiting for the second referee report before posting our responses and an updated draft. However, given the delay of the second report we are now already posting our responses to the first report below. The fully updated manuscript will be posted once the second report is submitted, but we have already included the envisaged text changes in our answers to the referee’s queries below.

With best wishes, The Authors

Requested changes:

Minor changes / questions:

1- The conditioning in Eq. 21 is not clear. The weight min($s$,$s_{max}$) in Eq. 19 originates from the probability of belonging to the accepted subset (where overweights are always accepted and thus saturate to 1) but Eq. 21 seems to consider only the non-overweights which have weight $s$, and not the overweights. I assumed this is because for small enough $\epsilon$, overweights need not be considered (same goes for Eq. 20 and as indicated in footnote 3). Could the authors clarify whether this is the case?

Yes, this is the case: “The approximation assumes small enough $\epsilon$, such that min($s$,$s_{max}$)$\approx s$ and min($x$,$x_{max}$)$\approx x$ can be assumed, when averaging.” We added this sentence for clarity.

2- Since the modification introduced in Eq. 16 is shown to be effective in enhancing the performance, it would benefit from a more detailed explanation as to why it works. The figure and the related discussion already show that this is because overweights are kept more or less constant while the second step efficiency increases, but the authors could clarify why this is the case specially because the notation in the cited master thesis is slightly different (and $x$ is not re-defined, just the weight at least in the version found here https://amslaurea.unibo.it/id/eprint/28767/1/Improving_MadGraph5.pdf).

One can better understand the difference with a graphic which we would like to show in Ref. [46]. Appended, you will find a sketch of this graphic. In the appended sketch, $s_{max}$ and $x_{max}$ are shown in the $s$-$w$ plane. Events are expected to be distributed around the main diagonal. $s_{max}$ is in both cases a vertical line. For the old definition $x_{max}$ is a diagonal going through the origin of coordinates. For the new definition $x_{max}$ gets horizontal when crossing $s_{max}$. Events below $x_{max}$ get rejection sampled while events above $x_{max}$ get an overweight. This means for $s$ above $s_{max}$ more events are kept with the new definition, improving the second unweighting efficiency. Especially, in the triangle on the top right the differences become clear: With the old algorithm events might be rejected there in the second unweighting step, but will always have a weight larger than 1 if kept. With the new algorithm these events are always kept with a lower final overweight, improving the efficiency. For an apple to apple comparison $x_{max}$ might need to be lowered for the new definition to get the same overweight fraction as stated in the legend of Figure 1. For $\epsilon_s=1$ the difference is most extreme. The old algorithm will use an $x_{max}$ similar to the shown black line to achieve $\epsilon_x=0.001$. All kept events will have an overweight from the first unweighting step. To have the same overweight fraction, the new $x_{max}$ has to be on the bottom, such that each event gets an overweight from the second unweighting step.

3- Additionally, how is s computed for Fig. 1? Is it with the same surrogate model or with two surrogate models trained with different definitions of $x$?

To clarify this we are going to add: “The efficiencies resulting from the same surrogate model are displayed as a function of the first unweighting maximum reduction [...]” in the caption of Figure 1. It is justified to use the same loss function during training for the comparison, as the approximations made in Eq. 35 drop the difference between the new and the old definition.

4- p.16 Eq. 34. Here, $\langle t_{surr}\rangle$ is assumed to be fixed and consequently not optimized. For a fixed architecture, this is clearly correct. However, there is an optimization of architecture described in 3.3. How does the change in architecture impact $\langle t_{surr}\rangle$? I assume it is a minor effect, but the authors should comment on this.

$\langle t_{surr}\rangle$ varies less than a factor 1.5 for the architectures we considered. It only enters the effective gain factor summed with $\langle t_{PS}\rangle$, which is a factor 20 larger than the nominal $\langle t_{surr}\rangle$, see Table 5. Thus, these architectural changes impact the final effective gain factor by less than 5% only. Nevertheless, this might provide a slight tendency to pick hyperparameters values with less complexity if the estimated gain factors are similar.

5- p.16 Below Eq. 35, $\langle x\rangle_s \approx \frac{\langle w\rangle}{\langle s\rangle}$ assumes $w$ and $s$ are independent. While I understand this is an empirical assumption meant to facilitate the optimization of $s$ with a suitable loss function, $w$ and $s$ should be correlated for a good surrogate model. Are there any alternatives to this assumption as for example using $\langle x\rangle(\theta)$ or is it too unstable?

This needs to be clarified in the paper: in the old definition of x this is not an approximation: $\langle x\rangle_s = \langle \frac{w}{s}\rangle_s = \frac{\langle \frac{w}{s}\cdot s \rangle}{\langle s\rangle}=\frac{\langle w \rangle}{\langle s\rangle}$. However, with the new definition there is an approximation: $\langle x\rangle_s = \langle\frac{w}{s}\cdot max(1, \frac{s}{s_{max}})\rangle_s \approx \langle \frac{w}{s}\rangle_s$. In either case Eq. 35 would only assume independence if $x$ would not be weighted with $s$. Using directly $\langle x\rangle_s$ is unfavourable for the back propagation because of the “max(.)” which discards the $s$-information from events with $s>s_{max}$. We add in the paper for clarification “$\langle x\rangle_s\approx \langle \frac{w}{s}\rangle_s=\frac{\langle w \rangle}{\langle s\rangle}$”.

6- p.17 Below Eq. 36, the authors observe that a randomly initialised network achieves lower gain factors than using MSE loss. Could the authors expand on this? Is this some pathology related to the fact that, for gain purposes, the overall scale of x is not important?

We clarified this sentence: “We observe that using this loss function for training directly from the start achieves a lower gain factor than training first with the MSE loss and afterwards with this loss function.” The gain factor loss function mainly rearranges the mean $\langle s(w)\rangle$ (see $w$-$s$ plane in answer 2 as motivation), while the MSE loss reduces the variance of s(w). Reducing first the variance and afterwards the mean turned out to be most effective.

Very minor changes / questions:

1- p. 3 “unit-weight events [...] are needed” This is explained in Sec 2.5, but perhaps a simple “unit-weight events [...] are preferred to avoid statistical dilution due to increased variance” (if this is what the authors mean) would increase clarity.

Thank you. We adjusted this sentence as you proposed.

2- p. 8 When introducing biased phase space sampling, I would add below Eq. 9 that even if certain phase-space configurations are enhanced, the final sample is unbiased, and reference Eq. 27.

To clarify and highlight this important fact, we have added a sentence right after Eq. (9): “Note that in order to finally obtain unbiased samples of unweighted events the enhance factor $h$ must be properly taken into account in the unweighting procedure, cf. Sec. 2.7.”

3- Ref. 47 is cited as the original source of Eq. 16. Perhaps the authors should include a link if possible.

We added the url “http://amslaurea.unibo.it/28767/” as reference in the bibliography.

4- The end of 2.5 justifies Eq. 16 by showing how the fraction of overweights can be kept constant while ensuring higher efficiencies for the second step. This becomes important when exploring the gain factor in Eq. 33, maybe it would be better to highlight this with a brief reference to said equation at the end?

We thank the referee for this suggestion and added a sentence including a reference to Eq. 16 to the end of Sec. 3.1. “We note that the definition of $x$ with an overweight correction factor in Eq. (16) leads to a higher second unweighting efficiency (for the same total overweight contribution) and therefore helps to achieve large gains. The gain factor as defined in Eq. (33) will [...]”

5- Training time is not included in the gain factor, even if it’s included in the overall timing evaluation in section 4. I assumed this is because the training time itself, whatever it ends up being, will be very minor compared to generation time (similarly to the pre-running used to optimize integration). Is this a correct interpretation?

The reason is that the gain factor is independent of the final number of generated events. It is a measure of the potential of the trained surrogate. It is defined as the factor by which surrogate based event generation is faster than the baseline approach once the surrogate is already there. For a final apple to apple comparison the surrogate training time needs to be considered as done in section 4.- once the final number of needed events is known. Consequently, for a very small number of generated events dipole-ml-surrogates will be slower. However, for typical production campaigns the relative contribution of network training time or similarly the efforts to optimise the integrator will be subdominant.

6- The gain factor loss seems pretty unstable even after regularization, especially in validation. Could the authors comment on this and whether it’s an issue for training? The authors could also highlight that suboptimal training is not necessarily a problem as long as the gain factor is large enough, since unbiased results are always achieved.

We added the following sentences to the end of section 3.4: “The aim of the training is to get a surrogate that saves most computational time on an independent test data set. Due to the unbiased two step unweighting procedure, potential overtraining or fluctuations of the validation loss, as seen in Figure 2, is not disadvantageous.“

7- p.22 section 4.1.2, “ Having more events in Monte Carlo samples than expected in real data is a requirement by most modern analyses, as they rely on machine-learning based discriminators which need to be trained with an abundance of signal and background events.” This is true also for non ML-based analysis using “traditional” variables, and it is simply to reduce the MC-stats systematics in any MC-based analysis. I highlight this because this renders the increase in efficiency due to surrogate modelling even more important.

We have put this statement into perspective by rephrasing as “[...] especially if they rely on machine-learning [...]”. While higher MC stats benefit every analysis, the impact on ML discriminators is typically largest and thus the MC statistics for the training is typically aimed to be a factor of up to 10 higher than without ML application.

8- p.30 Regarding the last sentence, could the authors comment on the impact, if any, of negative weights on partial unweighting and the use of surrogate models when going beyond LO computations? Is the correction from Eq. 3 to Eq. 16 equally applicable to negative event weights as detailed in Ref. 31?

In fact the application beyond the leading order requires only small adjustments to cover also negative weights, as they appear in NLO calculations. In Ref. [31] we have already proven that the surrogate unweighting algorithm can be generalised accordingly. To explicitly state this in the paper, we have added a small paragraph at the very end of the conclusions: “At the next-to-leading order event weights are no longer strictly positive. However, the surrogate-unweighting algorithm has already been shown to be generalisable to non-positive definite target distributions [31]. Its application for simulations beyond the leading order essentially requires to keep track of the proper event-weight sign and to employ the weight and surrogate modulus in the rejection sampling steps. “

Besides the revisions quoted above, in the overhaul of the manuscript we found and cured two inaccuracies in Eqs. (26), (27), where we replaced “$\langle w_{overweight} \rangle$” by “$\frac{\langle w \rangle}{\epsilon_{baseline} \cdot w_{max}}$”.

Attachment:

---

## Round 2 · Referee Report · Anonymous (Referee 2) · 2025-9-23

Strengths

  1. The authors convincingly demonstrate the feasibility and impact of surrogate unweighting for realistic Z+jets merged samples, including many subtleties of practical event generation.

  2. The paper provides clear performance benchmarks and illustrates how surrogates can yield order-of-magnitude speed-ups in relevant high-multiplicity channels.

  3. The methodological developments (gain-factor loss, staged overweighting) are novel and could be helpful beyond the specific case studied.

Weaknesses

  1. Equations (5), (14), (16), and (23) all define related overweight factors, but the connections only become clear after reading across several sections. A short roadmap paragraph would help readers follow the logic.

  2. Have the authors considered a fused expression, e.g. \begin{align} \tilde{w} =\max!\left( 1,\frac{s}{w_{\max}} \cdot \max!\left( 1,\; \frac{x}{x_{\max}} \right) \right) \end{align} which could partly compensate for the first-stage overweights automatically? How would this compare in efficiency and statistical dilution to the two-stage scheme? I believe a brief comment would be useful.

  3. Since this work involves practical modifications to Sherpa, I encourage the authors to comment on code availability. Will the surrogate unweighting machinery (training scripts, ONNX interfaces) be released or integrated into Sherpa in the future?

Report

I thank the authors for this very interesting and carefully presented manuscript. The topic is of high relevance for HL-LHC computing, and the paper represents a significant advance in the realistic deployment of surrogate unweighting in multijet-merged setups. The manuscript is generally well written, detailed, and technically solid. I recommend the manuscript for publication after minor revisions to address some clarifications (see details above).

Requested changes

See weaknesses

Recommendation

Ask for minor revision

  • validity: top
  • significance: top
  • originality: high
  • clarity: high
  • formatting: excellent
  • grammar: excellent

Author:  Steffen Schumann  on 2025-10-28  [id 5962]

(in reply to Report 2 on 2025-09-23)
Category:
remark

As requested by the editors we here attach the proposed revision of our paper, addressing the comments we obtained from the two referees.

Attachment:

MESurrogates2024_v2.pdf

Author:  Tim Herrmann  on 2025-10-21  [id 5948]

(in reply to Report 2 on 2025-09-23)

We thank the referee for the thorough reading of our manuscript and the very helpful additional comments, which further improve the paper on top of the first referee's comments. The fully updated manuscript will be posted once the editor agrees with our proposed changes. We included the envisaged text changes in our answers to the referee’s queries below.

With best wishes, The Authors

Requested changes:

Equations (5), (14), (16), and (23) all define related overweight factors, but the connections only become clear after reading across several sections. A short roadmap paragraph would help readers follow the logic.

We thank the referee for stating the missing clarity. Indeed, we defined the variables $x$,$ x_{\text{max}}$ and $\tilde{w}$ two times with different meanings. To resolve this confusion we changed $x$ to $\tilde{x}$ and $x_{\text{max}}$ to $\tilde{x}_{\text{max}}$ in Eqs. (3)-(5). Additionally, we renamed $\tilde{w}$ to $\tilde{x}_{\text{overweight}}$ in Eq. (5) to better distinguish it from Eq. (39). Furthermore, we added in the end of Sec 2.1: “There, in Eq. (16) we define an improved $x$ compared to $\tilde{x}$, which results in a different $x_{\text{overweight}}$ in Eq. (23). Additionally, the reduced maximum $s_{\text{max}}$ instead of $w_{\text{max}}$ is used there in the first unweighting step for further performance gains, visible when comparing Eqs. (2, 4) with Eq. (22).” Near the end of section 2.5 we added: “[...] Compared to the original overweight definition, Eq. (5), this leads to the simpler expression Eq. (23). [...]”

Have the authors considered a fused expression, e.g.

$$\tilde{w} = \max!\left(1,\frac{s}{w_{\text{max}}}\cdot\max!\left(1,\frac{x}{x_{\text{max}}}\right)\right)$$
which could partly compensate for the first-stage overweights automatically? How would this compare in efficiency and statistical dilution to the two-stage scheme? I believe a brief comment would be useful.

Thank you for this comment. We assume that the exclamation marks are byproducts of the markdown and imply no mathematical meaning. Indeed, one could summarize our two step unweighting with the formula:

$$ w_{\text{overweight}} = \max\left(1,\frac{(w/s)}{x_{\text{max}}}\cdot\max\left(1,\frac{s}{s_{\text{max}}}\right)\right) $$
where $x_{\text{max}}$ is derived from the distribution of $(w/s)⋅\max\left(1,\frac{s}{s_{\text{max}}}\right)$. Since it is not obvious that $x_{\text{max}}$ is defined like this we did choose to not write it like this in the paper. The proposed overweight expression $\tilde{w} = \max\left(1,\frac{s}{w_{\text{max}}}\cdot\max\left(1,\frac{x}{x_{\text{max}}}\right)\right)$ implies that $\tilde{x}=\frac{w}{s}$, including the time expensive $w$, is used in the first unweighting step, which is by definition less efficient than the one-step unweighting when setting it up for the same statistical dilution.

Since this work involves practical modifications to Sherpa, I encourage the authors to comment on code availability. Will the surrogate unweighting machinery (training scripts, ONNX interfaces) be released or integrated into Sherpa in the future?

This is a very good point. We add the following sentence at the end of the conclusion: “While the implementation for next-to-leading-order event generation is work in progress, we plan to provide the surrogate unweighting functionalities for tree-level processes with one of the next Sherpa feature releases.”

---

## Round 3 · List of Changes

The changes correspond to the replies to the referee report. Details can be found there.

---

## Editorial Decision

refereeing_in_preparation